# CL-TAD: A Contrastive-Learning-Based Method for Time Series Anomaly Detection

**Huynh Cong Viet Ngu and Keon Myung Lee *** 

Department of Computer Science, Chungbuk National University, Cheongju 28644, Republic of Korea;
huynhcongvietngu1.1@gmail.com
* Correspondence: kmlee@cbnu.ac.kr

**Abstract:** Anomaly detection has gained increasing attention in recent years, but detecting anomalies in time series data remains challenging due to temporal dynamics, label scarcity, and data diversity in real-world applications. To address these challenges, we introduce a novel method for anomaly detection in time series data, called CL-TAD (Contrastive-Learning-based method for Times series Anomaly Detection), which employs a contrastive-learning-based representation learning technique. Inspired by the successes of reconstruction-based approaches and contrastive learning approaches, the proposed method seeks to leverage these approaches for time series anomaly detection. The CL-TAD method is comprised of two main components: positive sample generation and contrastive-learning-based representation learning. The former component generates positive samples by trying to reconstruct the original data from masked samples. These positive samples, in conjunction with the original data, serve as input for the contrastive-learning-based representation learning component. The representations of input original data and their masked data are used to detect anomalies later on. Experimental results have demonstrated that the CL-TAD method achieved the best performance on five datasets out of nine benchmark datasets over 10 other recent methods. By leveraging the reconstruction learning and contrastive learning techniques, our method offers a promising solution for effectively detecting anomalies in time series data by handling the issues raised by label scarcity and data diversity, delivering high performance.

**Keywords:** anomaly detection; time series; reconstruction-based learning; contrastive-based learning; representation learning



## 1. Introduction

Anomaly detection in time series data has been a significant research topic for a number of applications such as intrusion detection in cybersecurity [1], fraud detection in finance [2], and damage detection in manufacturing [3]. Anomalies in time series data can be classified into point-wise and segment-wise anomalies [4]. Point-wise anomalies refer to unexpected events that occur at individual time points, typically having extreme values compared to the rest of the time points, or relatively deviated values from their neighboring points. In contrast, segment-wise anomalies typically are anomalous subsequences.

The time series anomaly detection task is to identify where a given time series (a sequence of real values) $X = (x_0, x_1, x_2, \ldots, x_i, \ldots, x_{T-1})$, where $T$ is the length of $X$, contains any anomalous events. This is commonly carried out by producing an output label sequence $Y = (y_0, y_1, y_2, \ldots, y_i, \ldots y_{T-1})$, where $y_i \in \{0, 1\}$ indicates whether its corresponding value $x_i$ is anomalous or not.

Various methods have been developed for the task, which can be categorized into five approaches: statistical [5–9], signal analysis (SA) [10–12], system-modeling-based [13], machine learning (ML) [14–19], and deep learning (DL) [20–43]. The statistical approach involves the creation of a statistical model by calculating distributions and measures such as mean, variance, median, quantile, and others. The SA-based approach utilizes

time–frequency domain analysis techniques like the Fourier transform, to identify anomalies. The system-modeling-based approach constructs a mathematical model such as a Bayesian dynamic linear model [13] that simulates the latent process to produce the time series of interest. The ML-based approach constructs a trained model for anomaly detection using unlabeled and/or labeled data. The DL-based approach, which has recently gained significant attention, makes use of DL models for anomaly detection, and has found widespread adoption, achieving numerous successes across diverse application domains. Despite the successes of DL models in various applications, the performance of DL-based anomaly methods in the time series domain has not yet been as convincing as expected. To date, DL-based anomaly methods still suffer from the data scarcity problem caused by rare anomalous events and rare labeled data.

Instead of using supervised learning techniques based on labeled data, the DL-based anomaly methods usually employ one of three unsupervised or self-supervised techniques: forecasting-based, reconstruction-based, or one-class classifier techniques. Forecasting-based techniques involve forecasting the value for the next time point from the preceding data. Any significant deviation between the predicted and actual value suggests an anomaly. Reconstruction-based techniques build a model to reconstruct the output similar to its input; discrepancies between the model's output and the actual data can indicate anomalies. One-class classifier techniques construct a classifier model for only normal data and leverage this model to determine if a given value of interest belongs to the normal class or not, thus enabling anomaly detection.

Contrastive learning on unlabeled data has emerged as an effective technique for learning representations useful to downstream tasks. The learning technique has shown potential in anomaly detection task for time series data [44–46], especially when available training data does not include anomaly events. The performance of contrastive learning is strongly affected by the approach used to generate positive and negative samples from available unlabeled data. In time series data anomaly detection, it is also challenging to find out an appropriate sample generation method.

Motivated by the successes of reconstruction-based learning techniques and the potential of contrastive learning for unlabeled data, we propose a novel method for anomaly detection in time-series data, called CL-TAD (Contrastive-Learning-based method for Times series Anomaly Detection). The key aspects of the proposed method are as follows:

- The CL-TAD method is comprised of two main components: positive sample generation and contrastive-learning-based representation learning. The first component plays role of generating positive samples from normal time series data. This is accomplished through a reconstruction process that tries to recover the original data from masked data. This reconstruction-based sampling is expected to augment a scarce original dataset into a robust and large training dataset that retains the essential information of normal patterns. The second component applies a contrastive learning technique to both the generated samples and the original data to produce their representations. Instead of directly producing the representations for generated samples, a transformation function with learnable parameters is applied to those samples to extract more meaningful information. The employed learnable transformation is attributed to learning more effective representations for anomaly detection.
- The CL-TAD method employs a temporal convolutional network (TCN) [47] for temporal feature extraction within time series data. The TCN is used in both positive sample generation and representation learning components.
- The proposed method uses a reconstruction-based data augmentation for the normal time series dataset to generate a sufficient amount of training data. On training the model for anomaly detection, it uses only the original data and their augmented data, and hence it is a nonsupervised learning approach that does not require the labeling task over the training data.
- To evaluate the effectiveness of our proposed method, we conducted some experiments on the following well-known nine benchmark datasets for time series anomaly detec-

tion: Power demand [21,27], ECG and 2D-Gesture [21,27], UCR and SMD [43], PSM [42], MSL [43], SWaT [42], and WADI [42]. The performance of the proposed method has been compared with those of 10 recently developed anomaly detection methods: MAD-GAN [32], DAGMM [22], MSCRED [36], CAE-M [28], OmiAnomaly [30], TranAD [42], GDN [29], Anomaly Transformer [42], MTAD-GAT [38], and USAD [39]. The experimental results have demonstrated that the proposed method achieved competitive performance on all of these datasets. Notably, the proposed method achieved the best performance on the following nine benchmark datasets: Power Demand, UCR, ECG, 2D-Gesture, PSM, SMD, MSL, SWaT, and WADI.

The remainder of the paper is organized as follows: Section 2 describes some related works on time series anomaly detection. Section 3 presents the proposed anomaly detection method in detail. In Section 4, we present the experimental results of the proposed method, comparing it with several recent anomaly detection methods using nine benchmark datasets. Finally, Section 5 draws conclusions about the proposed method.

## 2. Related Works

This section first briefly describes the deep-learning-based approaches to time series anomaly detection. Then it delves into how contrastive learning techniques can be used in time series anomaly detection. After that, it presents the temporal convolutional network (TCN) that is used in the proposed method.

### 2.1. Deep-Learning-Based Approaches to Time Series Anomaly Detection

As illustrated in Figure 1, the existing anomaly detection methods for time series data can be categorized into three primary approaches: one-class-classification-based, reconstruction-based, and prediction-based. One-class-classification-based methods employ a one-class classifier to identify anomalies. These methods define a latent hypersphere, where points falling inside the hypersphere are considered normal, while anomalies reside outside of it. The one-class classifier learns to differentiate between normal and anomalous data points, enabling the detection of anomalies in the time series data.

Reconstruction-based methods in time series anomaly detection often leverage the effectiveness of encoder–decoder models or their variants, which have achieved notable successes in solving reconstruction problems in computer vision. These methods aim to reconstruct the input time series data using an encoder–decoder architecture, where the quality of the reconstructed output serves as an indicator of anomalies. That is, the deviations or discrepancies of the input and reconstructed data can be used to identify anomalies, the larger difference, the higher the possibility of anomalies at the corresponding time points.

Lastly, prediction-based methods aim to forecast future values by training a predictive model. The predicted values are then compared with the actual values, allowing the identification of anomalies based on the discrepancies between them. By detecting significant deviations between the predicted and actual values, anomalies in the time series can be identified.

In both reconstruction-based and prediction-based methods, a predefined threshold, denoted as $\delta$, is commonly employed to determine whether a specific time point is anomalous. During the inference phase to identify anomaly events, the label $\hat{y}_t$ of a new data point at time step $t$ is assigned to an anomaly if the corresponding anomaly score $S(\hat{y}_t)$ exceeds the threshold $\delta$, as described in Equation (1).

$$\hat{y}_t = \begin{cases} 1, & \text{if } S(\hat{y}_t) \geq \delta. \\ 0, & \text{otherwise.} \end{cases} \tag{1}$$

where $\hat{y}_t$ represents the label assigned to the $t$-th data point, $S(.)$ quantifies the degree of abnormality or deviation of a data point from the normal patterns, and $\delta$ is a prespecified threshold.

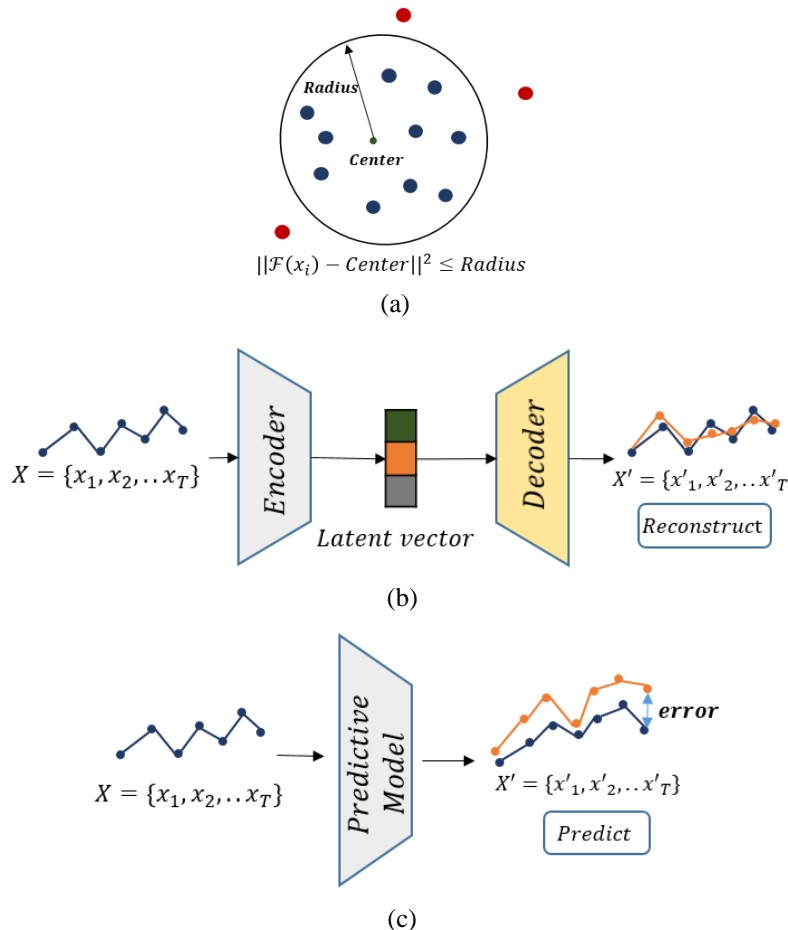

**Figure 1.** Deep-learning-based approaches for anomaly detection in time series. (**a**) One-class-classification-based approach; (**b**) Reconstruction-based approach; (**c**) Prediction-based approach.

DL-based approaches have significantly pushed up performance levels, yet they still fall short of the benchmark required for deployment in various real-world applications. One key reason for this unsatisfactory performance lies in the scarcity of time series training data. In time series anomaly detection, the quantity of time series data illustrating normal patterns is sometimes insufficient, let alone the severe lack of data representing anomalous events.

To tackle the issue of data scarcity, data augmentation techniques have been employed in deep learning, particularly in computer vision tasks. A reconstruction-based approach can leverage a masking technique to augment the training data available for building a reconstruction model. Deep-FIB [21] exemplifies such a method; it trains a reconstruction model using masked time series data. As illustrated in Figure 2, this method utilizes an encoder–decoder model to reconstruct the original time series from its masked series. Once such a model is trained, it is used to reconstruct the given input and then the reconstructed value is compared with its actual input to determine its anomaly. In the reconstruction-based approach, the masking strategy is important in building a model to extract meaningful characteristics for reconstruction. There are two main issues to be considered in the masking strategy: The first issue involves determining the amount of masking portion for input time series. Either overmasking or undermasking can cause a negative effect on the reconstruction capability of the model. Striking the right balance is crucial to ensure high performance. The second issue is about the masking value to be filled in the masked locations. Normalization of input time series sometimes needs to be applied. This normalization can affect the masking value for the normalized time series data.

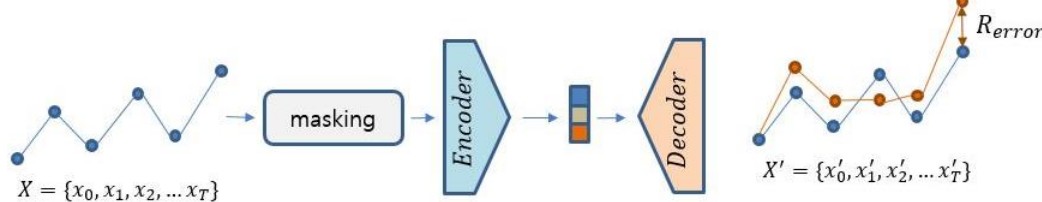

**Figure 2.** Deep-FIB method for time series anomaly detection.

## 2.2. Contrastive Learning for Unlabeled Data

Contrastive learning is a self-supervised approach that can be used for representation learning of unlabeled data. An early application of contrastive learning was to learn representations of image data by contrasting similar and dissimilar samples [48]. Figure 3 shows how a contrastive learning method usually works for image data: The method begins with generating positive images by applying predefined transformation functions $T$ to the original images in the initial dataset. Typical operations for transformation include color jitter, flipping, and rotation, as these operations are supposed to preserve the characteristic information about the image label. Negative samples are chosen randomly from the original image dataset, which are different from the anchor image. An encoder module $E$ is subsequently trained to make the representations of pairs of an anchor data sample and a positive sample similar while creating a substantial difference between the representations of pairs of an anchor data sample and a negative sample. However, in the time series domain, finding suitable transformation functions for all types of time series data is challenging as the transformation functions need to preserve the intrinsic temporal properties of the data.

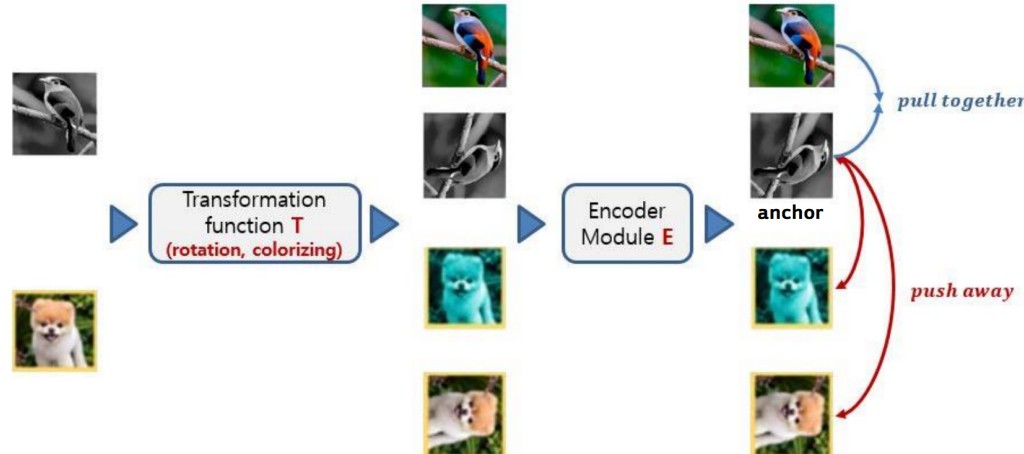

**Figure 3.** Basic contrastive learning scheme in the visual domain. First, transformation functions are utilized to generate positive pairs. Subsequently, the encoder is trained to draw the representations of similar pairs closer together, while pushing away the representations of pairs consisting of dissimilar ones.

Inspired by contrastive learning techniques that have been successful in computer vision, several contrastive learning methods have been proposed for time series analysis tasks (anomaly detection task [44–46,49] , classification task [50], and forecasting task [51]). In time series anomaly detection tasks, contrastive-learning-based models are trained typically on a normal dataset that does not include anomaly events. They might differ in how to obtain similar (positive) and dissimilar (negative) samples from the dataset. It is important to use a sampling method appropriate to the application domain.

TS2Vec [44] employs a contrastive-learning-based technique to obtain effective representations for anomaly detection. It randomly samples fixed-length subsequences from a time series dataset and projects them into a latent space using a learnable affine transforma-

tion. The projected vectors are then randomly masked and fed to a TCN (temporal convolutional network) [47] to produce their representations. TS2Vec samples the overlapped subsequences and considers the adjacent subsequences as positive pairs. Additionally, it considers pairs of samples, generated by applying different transformation functions such as jitter, scaling, and permutation to a subsequence, as positive samples. Conversely, it treats as negative samples, the non-overlapped and non-adjacent samples from a time series instance and the pairs of subsequences from different time series instances. It trains an encoder model to produce the representations in a way that increases the similarities of the representations for anchor and positive pairs and decreases those of the representations of anchor and negative pairs. Experiments showed that the performance of representation learning is influenced by the employed transformation functions.

The COCA method [45] builds a one-class classifier model for time series anomaly detection. It uses a contrastive learning technique to produce a latent space representation for normal data. To augment a training dataset, it applies transformation functions like jitter and scaling, the hyperparameters of which should be carefully chosen. It trains a TCN and LSTM-based encoder–decoder model to produce representations of time series data. On training the model, the pairs of original time series and their transformed series are used as positive pairs to enforce their latent representations to be sufficiently close with a contrastive learning technique. The COCA model is trained to minimize both the reconstruction loss of the encoder–decoder model and increase the similarities among the representations of the positive samples. It does not use negative pairs to train the model. The model is basically a deep SVDD model for a one-class classifier, and hence, it inherits the constraints of SVDD models.

Neutral AD [46] is another contrastive-learning-based method used for anomaly detection in time series data. It employs learnable neural transformations to augment time series data instead of predefined transformations like jitter and scale. The transformation is implemented by a trainable feed-forward neural network. The original samples and their transformed samples are then fed into an encoder to map them into a latent space. The transformed samples are regarded as positive samples to their original sample, while all other transformed samples are regarded as negative samples. The parameters of both the transformation network and the encoder are trained to maximize the similarities between an original sample and a positive sample and minimize those between an original sample and a negative sample. The developers of Neutral AD have claimed that the learnable transformations of original data are valuable for data augmentation in time series applications.

In reconstruction-based anomaly detection for time series data, having an effective tool for augmenting training data is crucial. Proper sampling and transformation techniques play a significant role in enhancing the performance of time series anomaly detection. In our proposed method, we introduce a novel strategy to address this challenge and improve the overall process.

### 2.3. Temporal Convolution Network

Motivated by the successes of CNNs in high-level feature extraction for image data, 1D-CNNs have been employed in the time series domain [20,21]. It has been observed that 1D-CNNs are not good enough to extract long-dependency features [47]. To address this issue, the Temporal Convolutional Neural Network (TCN) [47] has been introduced; it is shown in Figure 4. A TCN model consists of stacked residual blocks, each of which is a block of dilated and causal 1D convolutional layers with a skip connection. In a TCN model, higher layers have large dilation steps, so that higher layers receive more global features than lower layers. It is worth noting that TCN uses causal 1D convolutions that do not refer to future information. Our proposed method uses the TCN architecture to extract features from time series data.

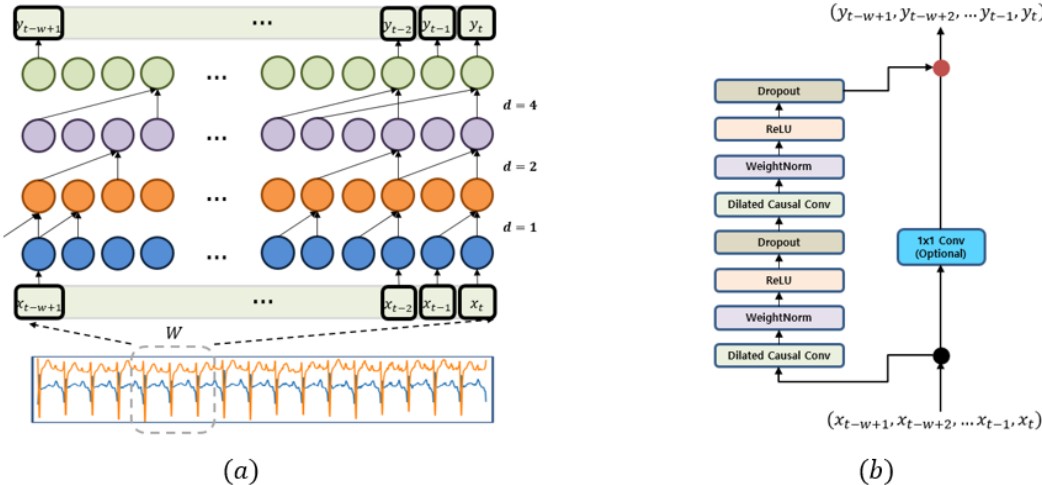

**Figure 4.** A TCN architecture for feature extraction of time series data: (**a**) shows hierarchical 1D convolutions, which are used to extract long-dependency features from the data. (**b**) displays the building block of a TCN architecture, which is comprised of residual, causal dilated convolutional blocks.

## 3. The Proposed Method for Time Series Anomaly Detection: CL-TAD

This section starts with the problem definition of interest in time series anomaly detection. It then provides an overview of the architecture of our proposed method and presents its components in detail. Finally, it describes the training and testing strategy employed by the method.

### 3.1. Problem Definition

A time series data can be expressed in a sequence $X = (x_0, x_1, \ldots, x_i, \ldots, x_{T-1})$ of length $T$, where $x_i \in R^d$ is the $d$-dimensional data point at time step $i$. If $d = 1$, this indicates that $X$ is univariate, whereas if $d > 1$, this implies that $X$ is multivariate. The time series anomaly detection task aims to produce a new output sequence $Y = (y_0, y_1, \ldots, y_i, \ldots, y_{T-1})$, where $y_i \in \{0, 1\}$ indicates whether the corresponding input value $x_i$ is an anomaly event. This means that we are concerned with point anomalies in time series data.

Similarly to many other deep-learning-based methods, we create a training dataset comprising subsequence samples $W$ of length $L$ from the time series data $X$. This is achieved by sliding a window of size $L$ over $X$ with a step size of 1, as depicted in Figure 5. The generated sequence samples are denoted by $W = (W_0, W_1, \ldots, W_{L-T})$, where $L$ is the length of samples and $T$ is the length of the input time series data.

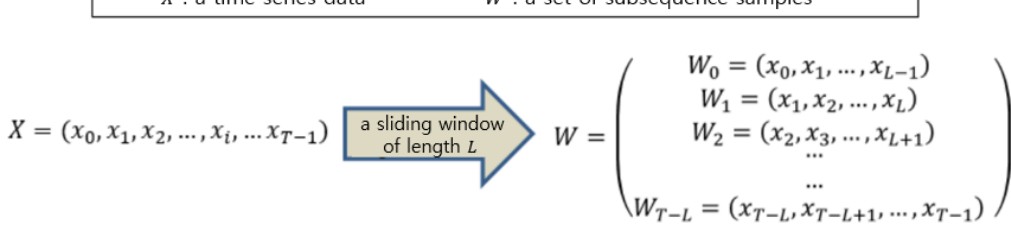

**Figure 5.** Generation of a set of subsequence samples $W$ from a time series data $X$: the subsequence samples are generated by copying the overlapped segments while sliding a window of length $L$ with the step size 1 over the given time series data $X$.

### 3.2. The Architecture for Representation Learning of CL-TAD

We propose a novel contrastive-learning-based anomaly detection method for time series data, which first augments the training data with an encoder–decoder-based reconstruction module using randomly masked samples, and then trains a contrastive-learning-

based representation learning module using those samples and original data. For anomaly detection, the anomaly scores are evaluated by calculating the differences between the representations of the original data with those of their corresponding reconstructed data. We refer to this proposed method as CL-TAD (Contrastive-Learning-based Time series Anomaly Detection).

CL-TAD is conducted in two sequential stages: the representation learning stage and the anomaly scoring stage. In the representation learning stage, latent representations for fixed-length subsequences of time series and their augmented subsequences are generated by a contrastive-learning-based representation learning module. In the anomaly scoring phase, the anomaly score for each time step is calculated by comparing the representations of two subsequences: The first is the original subsequence that ends at the time step, and the second is the subsequence that is reconstructed from the first subsequence of which the last time step is masked. Figure 6 illustrates the architecture used for representation learning in CL-TAD, which consists of two main components: reconstruction-based positive sample generation and contrastive-learning-based representation learning.

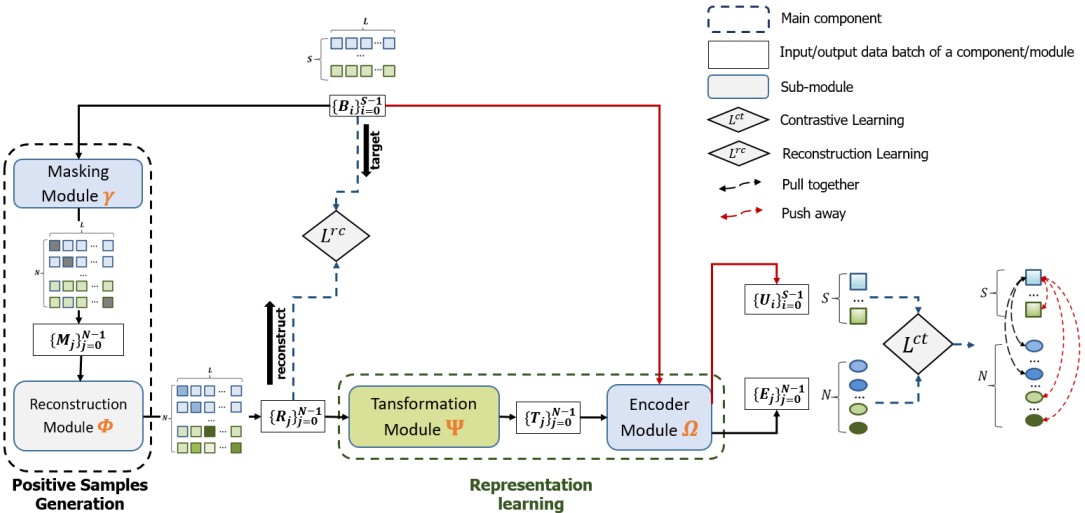

**Figure 6. CL-TAD**: Contrastive-Learning-based model for Time series Anomaly Detection.

The first component plays the role of generating positive samples by leveraging a reconstruction-based encoder–decoder network, which recovers masked samples of the original time series data. Its reconstruction-based learning approach with a masking strategy facilitates the generalization of a robust and large positive training dataset. The representation learning component then uses both the created positive samples and the original samples for training. The whole network for sample generation and representation generation is trained in an end-to-end manner.

Some contrastive-learning-based methods [44,45] employ pre-defined transformation functions to augment original time series data. Conversely, a method discussed in [46] adopts a learnable transformation function for the same purpose. In our approach, we integrate a learnable transformation module that pre-processes the augmented data prior to feeding them into the contrastive learning module.

### 3.3. Generation of Positive Samples

The component for positive sample generation is designed to create positive samples corresponding to each input subsequence sample. It consists of the masking module $\gamma$ and the reconstruction-based encoder–decoder module $\Phi$. From a given time series data $X$, the set $W$ of the original subsequence samples is generated, as depicted in Figure 5. In each learning iteration of the component, we first randomly choose from $X$ a batch of $S$ subsequence samples, each having a length of $L$. This batch is represented as $B = \{B_i\}_{i=0}^{S-1}$.

The masking module $\gamma$ produces a new batch of $N$ masked samples, $M = \{M_j\}_{j=0}^{N-1}$ where $N = S \times L$, from the batch $B$. For each $i$-th subsequence of length $L$ in $\{B_i\}_{i=0}^{S-1}$, a collection of $L$ masked samples is created by masking the value at each time step individually. Let $M_{j=iL+k}$ denote the data produced by masking the value at the $k$-th time step of the $i$-th sampled sequence $B_i$.

Figure 7 illustrates the process of creating a batch of 10 masked data, $\{M_j\}_{j=0}^{N-1}$, from a batch of two subsequence samples $\{B_1, B_2\}$, each with a length of $L = 5$. Please refer to Algorithm A1 in Appendix A for the detailed implementation.

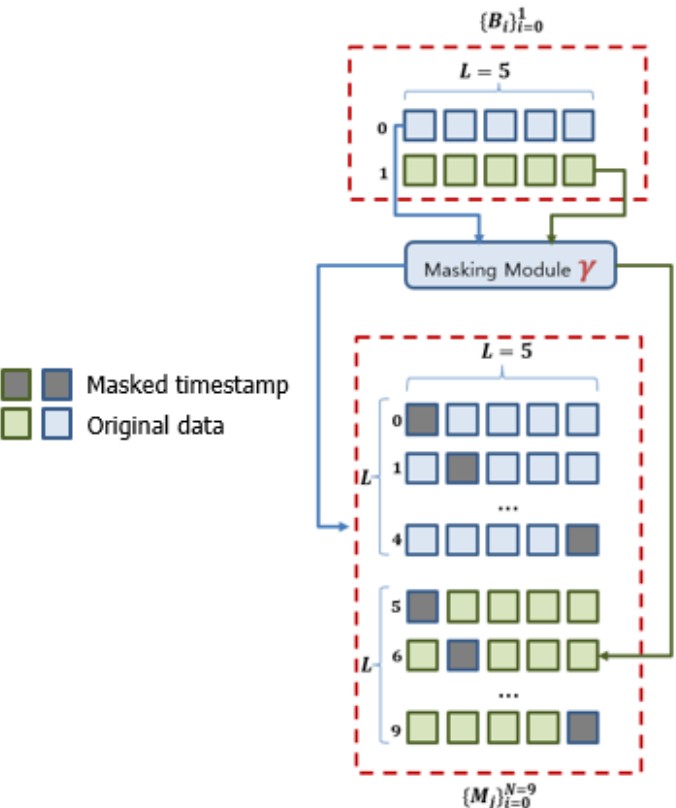

**Figure 7.** Generation of 10 masked samples $\{M_j\}_{i=0}^{9}$ from two original subsequence samples $\{B_1, B_2\}$ of length 5. Masked data are generated by masking the value at each time step one at a time.

The batch of generated masked samples $\{M_j\}_{j=0}^{N-1}$ is then fed into the module $\Phi$ for the reconstruction of its original sample, as shown in Figure 8. For the reconstruction task, the module $\Phi$ uses an encoder–decoder architecture to reconstruct original input samples $B_{\lfloor j/L \rfloor}$ from masked samples $M_j$. Specifically, the batch of masked samples $\{M_j\}_{j=0}^{N-1}$ is fed into the module $\Phi$ to generate reconstructed samples $\{R_j\}_{j=0}^{N-1}$, each of which is expected to be similar to $B_{\lfloor j/L \rfloor}$. In the module $\Phi$, we use a TCN network for the encoder and a linear layer for the decoder.

To train the module $\Phi$, we use the average difference between the reconstructed samples and their original samples as the loss function $L^{rc}$, as shown in Equation (2). Please refer to Algorithm A2 in Appendix A for a detailed implementation of the reconstruction learning procedure.

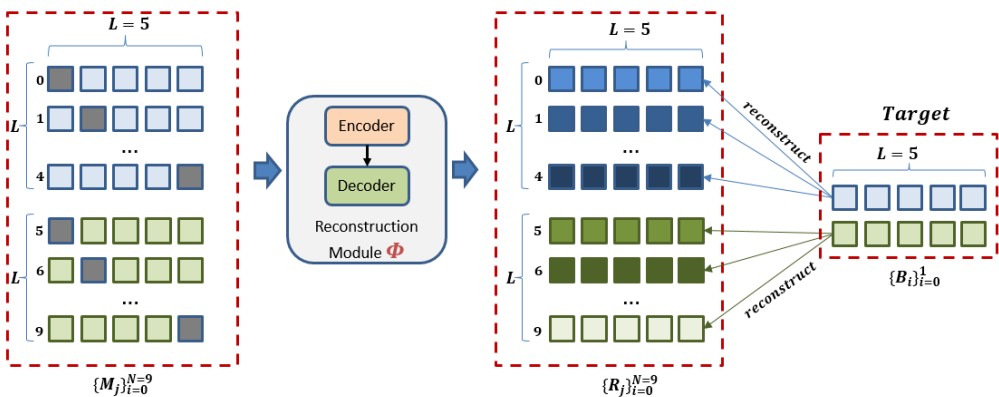

**Figure 8.** Illustration of reconstruction learning with module $\Phi$.

$$L^{rc} = \frac{1}{N} \sum_{j=0}^{N-1} \frac{1}{L} \sum_{t=0}^{L-1} ||B_{\lfloor j/L \rfloor}[t] - R_j[t]|| \tag{2}$$

where $N$ is the number of output samples, $L$ is the length of each sample, $\lfloor . \rfloor$ represents the floor function, $R_j$ is the $j$-th sample in $\{R_j\}_{j=0}^{N-1}$, and $B_{\lfloor j/L \rfloor}$ is the original sample of the reconstructed sample $R_j$.

*3.4. Contrastive-Learning-Based Representation Learning*

To generate the representations for anomaly evaluation, the proposed method uses a contrastive-learning-based component that consists of a learnable transformation module $\psi$ and its following encoder module $\Omega$, as shown in Figure 9.

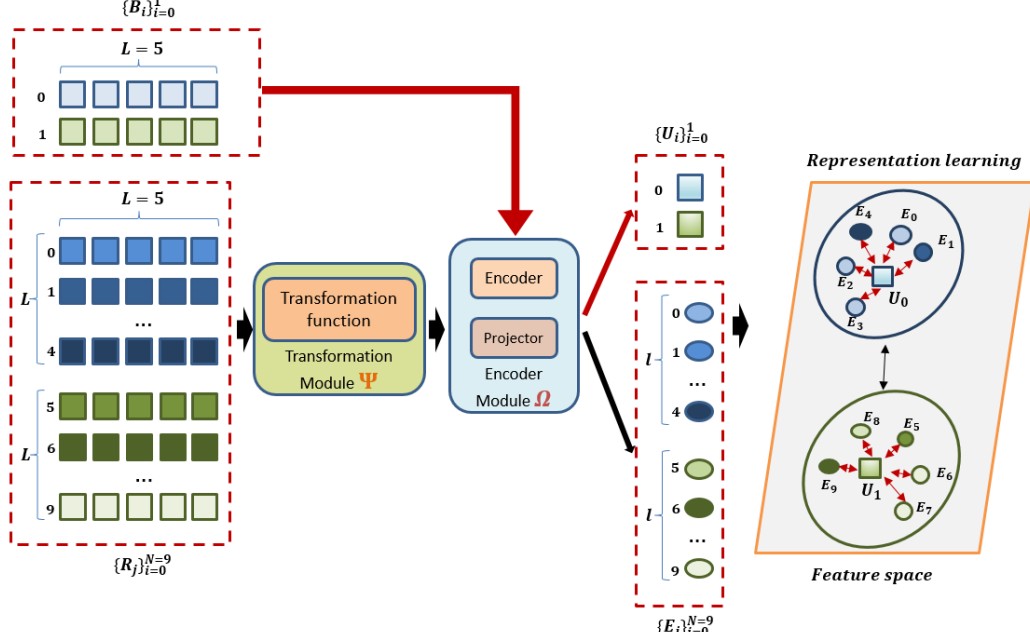

**Figure 9.** The architecture of the contrastive-learning-based representation learning module.

Both the original samples $\{B_i\}_{i=0}^{S-1}$ and the reconstructed samples $\{R_j\}_{j=0}^{N-1}$ serve as the training data for the contrastive-learning-based component. The original samples $\{B_i\}_{i=0}^{S-1}$ are processed directly by the encoder module $\Omega$. Conversely, the reconstructed samples $\{R_j\}_{j=0}^{N-1}$ go first through the transformation module $\Psi$, with their outputs $\{T_j\}_{j=0}^{N-1}$ subsequently being supplied to the encoder module $\Omega$. The encoder module consists

of an encoder and a projector. The input to the encoder module is first delivered to the encoder and then the output of the encoder is passed to the projector to create its corresponding representation. The representations of the original samples $\{B_i\}_{i=0}^{S-1}$ are denoted by $\{U_i\}_{i=0}^{S-1} = \Omega(\{B_i\}_{i=0}^{S-1})$, while those of the reconstructed samples $\{R_j\}_{j=0}^{N-1}$ are denoted by $\{E_j\}_{j=0}^{N-1} = \Omega(T_{j=0}^{N-1})$. In the representation learning component, we employ a linear layer network for the learnable transformation module $\Psi$, a TCN for the encoder of the encoder module $\Omega$, and a fully connected network for its projector.

The contrastive-learning-based representation learning component is trained to generate similar representations for pairs of an original sample and its corresponding similar sample (i.e., positive example), and dissimilar representations for pairs of an original sample and a different sample (i.e., negative example). For any given original sample, its associated reconstructed samples are regarded as positive examples, while both other original samples, as well as the reconstructed samples derived from them, are regarded as negative examples. For a sample $B_i$ within $\{B_i\}_{i=0}^{S-1}$, the representations of its corresponding reconstructed samples, denoted as $\{R_k^i\}_{k=iL}^{iL+L-1}$, thus form a set of $L$ positive examples associated with $B_i$. Meanwhile, the representations of the other samples $B_j, (j \neq i)$, as well as their reconstructed samples $\{R_k^j\}_{j \neq \{iL+L-1,...,k=iL\}}$, constitute a set of negative examples for $B_i$.

The representation learning component is trained to draw the representation, $U_i$, of $B_i$ and its positive examples $\{R_k^i\}_{k=iL}^{iL+L-1}$ closer, while simultaneously pushing the negative examples further away from $U_i$. To evaluate the similarity between two representations, **a** and **b**, we can use a similarity score function $z(\mathbf{a}, \mathbf{b})$ defined as follows:

$$z(\mathbf{a}, \mathbf{b}) = \exp\left(\frac{\mathbf{a}^T \mathbf{b}}{\tau \|\mathbf{a}\| \|\mathbf{b}\|}\right) \tag{3}$$

where **a** and **b** indicate vectorized representations for subsequence samples, while $\tau$ is the temperature hyperparameter that controls the sensitivity of the loss function in relation to the trained representations [52].

In practical applications, pinpointing an ideal value for $\tau$ could be complicated. To address this challenge, we integrate the TaU method [53] to dynamically adjust the $\tau$ parameter within the loss function. Previously proven effective in the visual domain, the TaU method offers a solution to this issue. Leveraging the capabilities of the TaU method, we propose a modified pairwise similarity score function, denoted as $v(\mathbf{a}, \mathbf{b})$, as follows:

$$v(\mathbf{a}, \mathbf{b}) = \exp\left(\frac{\mathbf{a}^T \mathbf{b}}{\|\mathbf{a}\| \|\mathbf{b}\|} * \frac{\sigma(u(\mathbf{a}))}{\tau}\right) \tag{4}$$

where $u(\mathbf{a})$ is the uncertainty parameter of **a** and $\sigma(.)$ denotes the sigmoid function. It should be noted that the uncertainty parameter $u(\mathbf{a})$ is fine-tuned during the representation learning phase.

Using the pairwise similarity score, the loss $L^{ct}(U_i, E_k)$ is computed between the representation $U_i$ of the original sample $B_i$ and the representation $E_k$ of a positive example $R_k$ from the set $\{R_k^i\}_{k=iL}^{iL+L-1}$ as follows:

$$L^{ct}(U_i, E_k) = -\log \frac{v(U_i, E_k)}{\sum_{j=0}^{N-1} 1_{[j \notin [iL, iL+L-1]]}[v(U_i, E_j)] + \sum_{m=0}^{S-1} 1_{[m \neq i]}[v(U_i, U_m)] + v(U_i, E_k)} \tag{5}$$

where $1_{cond}[val]$ signifies that *val* is used if the condition *cond* is satisfied; otherwise the value 0 is applied. $L^{ct}(U_i, E_k)$ indicates the contrastive loss associated with an original sample $B_i$.

Next, the loss $L^{ct}(E_j, U_h)$ is defined between the representation $E_j$ of a positive example $R_j$ and the representation $U_h$ of its associated original sample $B_h$, where $h = \lfloor j/L \rfloor$, as follows:

$$L^{ct}(E_j, U_h) = -\log \frac{v(E_j, U_h)}{\sum_{k=0}^{N-1} \mathbb{1}_{[k \notin [hL, hL+L-1]]}[v(E_j, E_k)] + \sum_{i=0}^{S-1} v(E_j, U_i)} \tag{6}$$

Finally, the average loss $L^{ct}$ over all pairs is computed as follows:

$$L^{ct} = \frac{1}{2}\left(\left[\frac{1}{N}\sum_{i=0}^{S-1}\sum_{k=iL}^{iL+L-1} L^{ct}(U_i, E_k)\right] + \left[\frac{1}{N}\sum_{j=0}^{N-1} L^{ct}(E_j, U_{\lfloor j/L \rfloor})\right]\right) \tag{7}$$

where the first term indicates the average loss between an input sample $U_i$ and its corresponding positive example $E_k$ and the second term indicates the average loss between a representation $E_j$ and its corresponding positive sample $U_h$.

### 3.5. Training of Representation Learning Model for the CL-TAD Method

In the representation generation stage, the CL-TAD method follows a two-step process. Firstly, it leverages the masking module combined with the reconstruction module $\Phi$ to create positive samples. Secondly, it generates the representations of these samples using the transformation module $\Psi$ and the encoder module $\Omega$. The modules $\Phi, \Psi,$ and $\Omega$, together with the loss function $L^{ct}$, encompass the learnable parameters. Those modules are implemented with neural network models.

These modules are jointly trained in an end-to-end manner using the combined loss function $L^{final}$. This function amalgamates the reconstruction loss $L^{rc}$ from Equation (2) with the contrastive learning loss $L^{ct}$ from Equation (7). Such integration facilitates holistic learning and optimization of the modules throughout the training process.

$$L^{final} = L^{rc} + L^{ct} \tag{8}$$

The CL-TAD method trains the representation learning model on a time series dataset without any abnormal events. Original samples are generated by applying a sliding window of length $L$ across the given time series data.

For a particular batch of samples, each sample undergoes the masking process using the module $\gamma$, resulting in masked samples. These masked samples are then reconstructed via the reconstruction module $\Psi$. The original samples directly enter the encoder module $\Omega$. In contrast, the reconstructed samples are first passed through the transformation module $\Psi$ before being processed by the encoder module $\Omega$. The subsequent original and reconstructed samples are paired to form positive and negative samples for contrastive learning.

The optimization of the model is then driven by the loss function $L^{final}$, which is computed based on these paired arrangements. Such a comprehensive training approach ensures that the model effectively learns and refines representations. For a detailed description of the training procedure, please refer to Algorithm A3 in Appendix A.

### 3.6. Anomaly Score Evaluation

For a given test time series data $Z = \{z_0, z_1, \ldots, z_S\}$, its anomaly score for each time step is computed to ascertain its potential status. The CL-TAD method employs a distance-based mechanism to determine these scores. The procedure begins by forming subsequences $B_i = (z_i, z_{i+1}, \ldots, z_{i+L-1})$ with a length of $L$ derived from the given data $Z$. This is achieved by shifting a window of length $L$ over $Z$. It is crucial to maintain this window length $L$ consistent with what was adopted during the training phase. To compute the anomaly score associated with the last time step of $B_i$, the value at the last time step is masked, yielding the masked subsequence denoted by $M_{iL+L-1}$.

The original subsequence $B_i$ is processed through the encoder module $\Omega$ to yield its representation $U_i$. In parallel, the masked subsequence $M_{iL+L-1}$ undergoes sequential pro-

cessing through the reconstruction module $\Phi$, transformation module $\Psi$, and the encoder module $\Omega$ to derive its representation $E_{iL+L-1}$.

The normalized Euclidean distance function is used to compute the distance $D(U_i, E_{iL+L-1})$ between the representation $U_i$ of the original subsequence $B_i$ and the representation $E_{iL+L-1}$ of the reconstructed subsequence $M_{iL+L-1}$ as follows:

$$D(U_i, E_{iL+L-1}) = \left\| \frac{U_i}{||U_i||} - \frac{E_{iL+L-1}}{||E_{iL+L-1}||} \right\| \tag{9}$$

The data point at the last time step in $W_i$, specifically $z_{i+L-1}$, is designated as anomalous only when the distance $D(U_i, E_{iL+L-1})$ exceeds the pre-specified threshold $\delta$. Figure 10 illustrates the process of generating and comparing the representations for the anomaly score calculation.

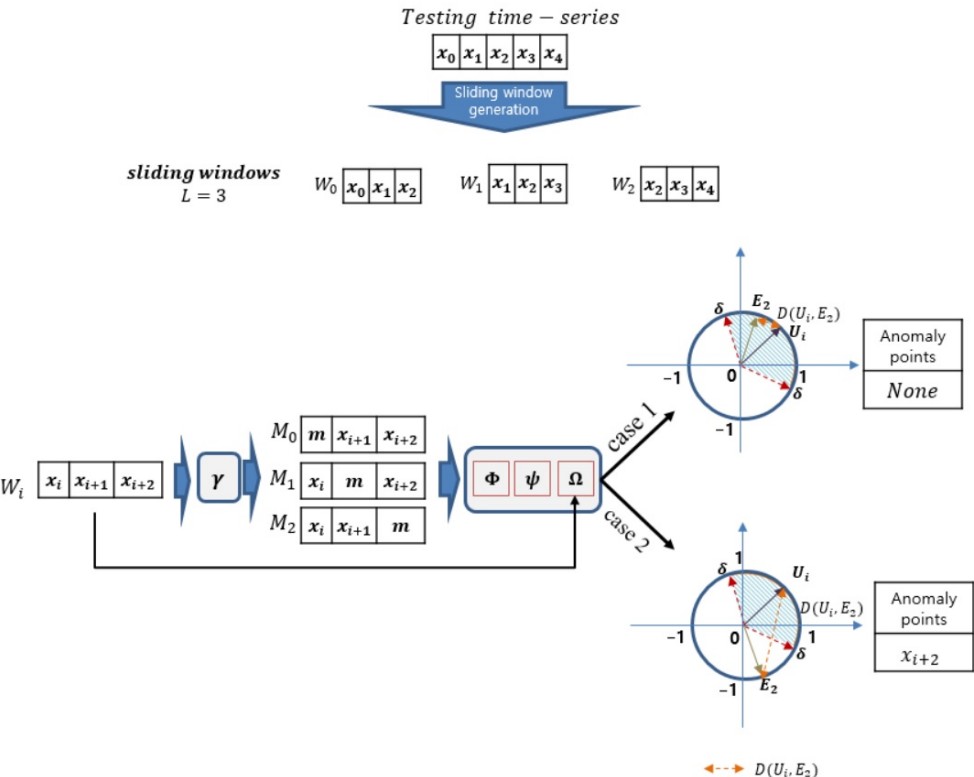

**Figure 10.** Representation generation and score computation for sequences of length 3. Each subsequence's final time step is masked to produce a corresponding masked subsequence, which is then reconstructed. The trained model transforms both the original subsequence and its reconstructed version into their respective representations. The difference between these two representations, as measured by their distance, provides the anomaly score.

The selection of the threshold value $\delta$ is based on the examination of the $F_1$ scores across a set of potential threshold values, uniformly distributed from 0 to the maximum representation distances observed on a held-out test dataset containing anomaly events. The value that produces the highest $F_1$ score is chosen as the threshold $\delta$. It is noteworthy that the distance distributions associated with anomalous events can differ depending on real-world applications. Algorithm A5 in Appendix A describes how the list of candidate thresholds is constructed.

This thresholding approach has been adopted in several anomaly detection methods, including RAMED [27], OmniAnomaly [30], DeepFIB [21], MAD-GAN [32], and others. The $F_1$ score is derived from precision and recall, and is defined as follows:

$$F_1 = \frac{2 \times Precision \times Recall}{Precision + Recall} \tag{10}$$

$$Precision = \frac{TruePositives}{TruePositives + FalsePositives} \tag{11}$$

$$Recall = \frac{TruePositives}{TruePositives + FalseNegatives} \tag{12}$$

Refer to Algorithm A5 in Appendix A used to generate the list of thresholds. For datasets consisting of multiple time series, the final performance is evaluated by computing the average $F_1$ scores over all sub-time series.

Moreover, the proposed method incorporates the point adjustment technique [11] after performing threshold-based anomaly determination. Following this technique, once an individual time step within an anomalous segment is identified as anomalous, the entire segment is then considered as anomalous. This modification has demonstrated its effectiveness in practical scenarios, as recognizing an anomaly at a specific time step typically signals an alarm, bringing attention to its corresponding entire anomalous window. The point adjustment technique is visually presented in Figure 11. In this figure, the *Anomaly label* represents the ground truth, with 1 signifying the presence of an anomaly event at a given time step and 0 denoting regular activity. The *Raw-predicted Output* displays the preliminary anomaly labels generated through threshold-based anomaly identification. Meanwhile, the *Adjusted Output* portrays the results after applying the point adjustment technique.

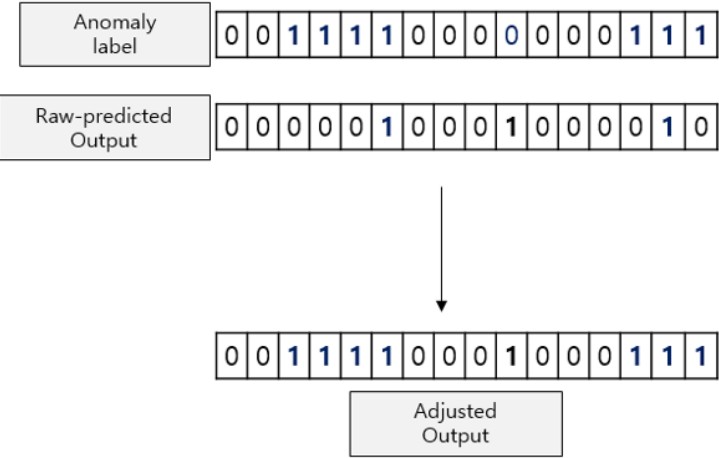

**Figure 11.** Illustration of point adjustment technique.

Algorithm A6 in Appendix A shows the entire procedure to conduct the best $F_1$-score-based performance evaluation.

## 4. Experiments and Discussions

This section outlines the results obtained from a series of experiments conducted to evaluate the effectiveness of our proposed model. These experiments were conducted on nine different datasets, widely recognized as benchmark datasets in the field. First, it briefly presents the dataset used for the experiments. Next, it describes how the proposed method has been implemented for the experiments. Subsequently, it shows the performance outcomes achieved by our method. Finally, it presents the outcomes of an ablation study.

### 4.1. Benchmark Datasets

The proposed method was tested on a combination of two univariate datasets—UCR and Power Demand—as well as seven multivariate datasets, namely ECG, 2D-Gesture, PSM, SMD, MSL, SWaT, and WADI. Power Demand [21,27,54] is a univariate dataset that records the yearly power demand at a Dutch research facility. The UCR (HexagolML)

dataset [43,55] was a part of the KDD-21 competition and serves as a univariate anomaly detection benchmark. It is divided into four subsets: 135_, 136_, 137_, and 138_.

The ECG dataset [21,27,54], consisting of two-dimensional electrocardiogram readings, is further segmented into six sub-problems: chfdb_chf01_275 (ECG-A), chfdb_chf13_45590 (ECG-B), chfdbchf15 (ECG-C), ltstdb_20221_43 (ECG-D), ltstdb_20321_240 (ECG-E), and mitdb_100_180 (ECG-F). The 2D-Gesture dataset [21,27,54] records two-dimensional X–Y coordinates of hand gestures captured in videos. The PSM dataset [42,56], with 25 dimensions, logs data from various application server nodes at eBay. The SMD dataset [43,57] is a 38-dimensional dataset capturing five weeks of resource utilization across 28 computers. Only the non-trivial traces labeled as *machine1-1, 2-1, 3-2, 3-7* were used in our experiments as per [43,57]. The MSL dataset [43,58], 55-dimensional in nature, comprises soil samples from the NASA Mars rover. Only the non-trivial trace, *C2*, was considered for anomaly detection, consistent with [43,58]. The SWaT dataset [42,59] has 51 dimensions and was gathered over a week of regular operations and four days of irregular operations at a real-world water treatment plant. Lastly, WADI [42,59] is an extended version of SWaT, boasting 127 dimensions, and encompasses data from a broader array of sensors and actuators. Table 1 presents the statistics of those datasets.

All datasets have been normalized to fit within the range [0, 1]. For datasets comprising multiple time series, each series underwent independent normalization. The normalization method is elaborated upon in Algorithm A7 located in Appendix A. Figure 12 offers a visual representation of the normalization for a time series from the two-dimensional ECG dataset. The green sections in the figure mark an anomalous event.

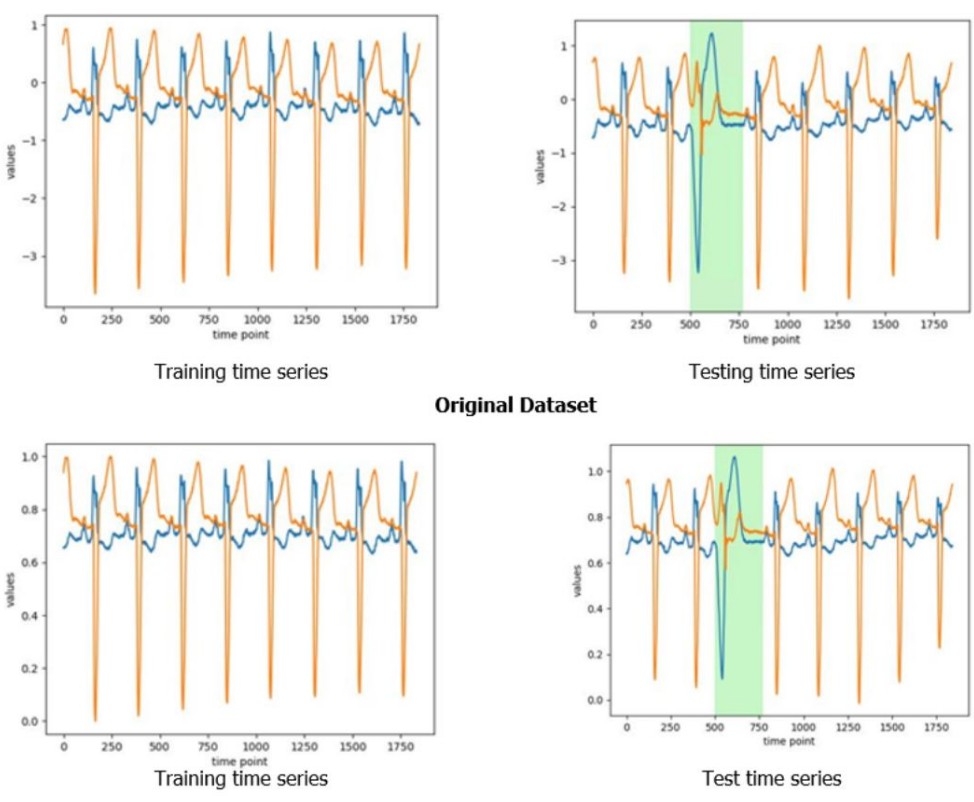

**Figure 12.** Normalization of a time series in the two-dimensional ECG data where each colored curve shows a trace of a signal. **Top left**: an original normal time series; **Top right**: an original time series with an anomaly; **Bottom left**: the corresponding normalized normal time series; **Bottom right**: the corresponding normalized time series with an anomaly.

**Table 1.** Statistics of the datasets used for evaluation.

| | Dataset | #Dimension | #Train | #Test | Anomaly (%) |
|---|---|---|---|---|---|
| | Power demand | 1 | 18,145 | 14,786 | 10.39% |
| UCR | 135_ | 1 | 1200 | 6301 | 0.19% |
| | 136_ | 1 | 1600 | 5900 | 1.88 |
| | 137_ | 1 | 2300 | 5200 | 1.96 |
| | 138_ | 1 | 3000 | 4500 | 0.22 |
| ECG | chfdb_chf01_275 (ECG-A) | 2 | 1833 | 1841 | 14.61% |
| | chfdb_chf13_45590 (ECG-B) | 2 | 2439 | 1287 | 12.35% |
| | chfdbchf15 (ECG-C) | 2 | 10,863 | 3348 | 4.45% |
| | ltstdb_20221_43 (ECG-D) | 2 | 2610 | 1121 | 11.51% |
| | ltstdb_20321_240 (ECG-E) | 2 | 2011 | 1447 | 9.61% |
| | mitdb__100_180 (ECG-F) | 2 | 2943 | 2255 | 8.38% |
| | 2D-Gesture | 2 | 8251 | 3000 | 24.63% |
| | PSM | 25 | 132,481 | 87,841 | 27.76% |
| SMD | machine1-1 | 38 | 28,479 | 27479 | 9.36% |
| | machine2-1 | 38 | 23,693 | 23,694 | 4.94% |
| | machine3-2 | 38 | 23,702 | 23,703 | 4.66% |
| | machine3-7 | 38 | 28,705 | 28,705 | 1.51% |
| MSL | C-2 | 55 | 764 | 2051 | 6.58% |
| | SWaT | 51 | 495,000 | 449,919 | 12.14% |
| | WADI | 127 | 1,209,601 | 172,801 | 5.7% |

### 4.2. The Implemented Model Architecture and Its Training

For anomaly detection on the benchmark datasets, we implemented a deep neural network model corresponding to the architecture shown in Figure 6. The reconstruction module $\Phi$ comprises an encoder, built using a six-layer TCN (temporal convolution network), and a decoder realized with a linear layer. For the representation learning component, the transformation module $\Psi$ was implemented with a linear layer. In contrast, the encoder of the encoder module $\Omega$ was implemented with a six-layer TCN, and the projector was realized with a two-layer fully connected network.

Table 2 presents the hyperparameter values used in the experiments. For the training of our designed network model, we employed the Adam [60] optimizer with the LARS wrapper [61]. To adjust the learning rate, we utilized the cosine annealing technique with a linear warmup of 10 epochs, starting with a learning rate of $10^{-3}$. Furthermore, the hyperparameter $\tau$ was set to 0.05. The entire model was implemented using Pytorch 2.0.1 and executed on an NVIDIA Geforce RTX 2080 Ti GPU.

**Table 2.** Training parameters.

| Hyperparameter | Set Value |
|---|---|
| Window size | 16 |
| Mask value | 0. |
| Batch size | 256 |
| Initial lr | 0.001 |
| lr scheduling | Consine annealing lr |
| Optimizer | Adam |
| tau ($\tau$) | 0.05 |

*4.3. Experiment Results and Comparison*

To gain insight into how the proposed method forms the representations of samples, the distribution of representations was visualized for some examples from the ECG and 2D-Gesture datasets. This was done for both pre-training and post-training, as depicted in Figures 13 and 14. To ensure understandable visualization, we employed the PCA technique to project these representations into a two-dimensional space. From the figures to show post-training results, it is clear that the positive samples cluster closely together while the negative samples are dispersed further away. This observation shows that our proposed method can generate meaningful representations.

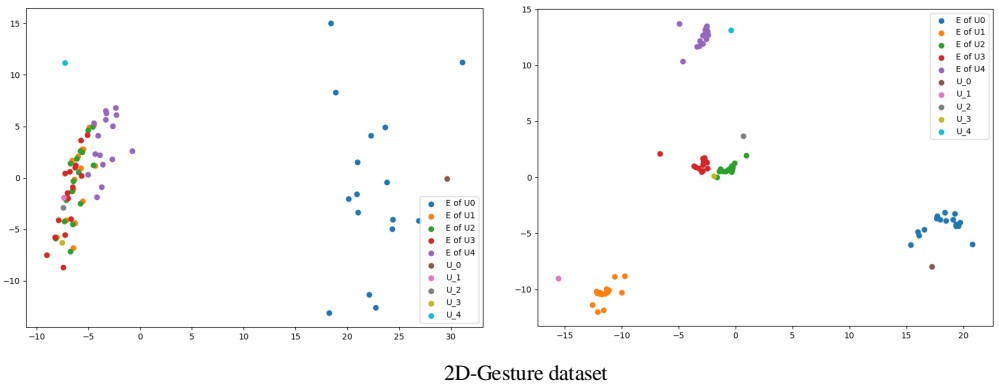

2D-Gesture dataset

**Figure 13.** Representation distributions before and after training for the 2D-Gesture dataset. **Left**: Pre-training; **Right**: Post-training.

Table 3 presents the experimental outcomes achieved by our proposed method on the benchmark datasets. We employed performance metrics such as precision, recall, and $F_1$ score for the evaluation. For datasets that are split into several sub-datasets, we have provided an in-depth performance breakdown for each sub-dataset in Tables A1–A3 located in Appendix B. For datasets such as ECG, UCR, and SMD, which are divided into multiple sub-datasets, the $F_1$ scores were averaged across all the sub-datasets to give a consolidated score.

To further show the effectiveness of our proposed method, we have compared its $F_1$ score performance with that of several state-of-the-art methods, as presented in Table 4, for the identical benchmark datasets. We made comparisons with the following methods: MAD-GAN [32], DAGMM [22], MSCRED [36], CAE-M [28], OmniAnomaly [30], TranAD [43], GDN [29], MTAD-GAT [38], and USAD [39]. The numbers in bold indicate the top-performing results for each benchmark dataset. Each compared method has achieved its best performance by using its curated optimal hyperparameters. We have utilized superscripts to indicate the source from which each performance score was derived: (1) refers to the model's original paper, (2) is sourced from [21], (3) from [42], (4) from [43], and (5) represents the performance scores that we obtained running the datasets through the

models using codes available from the developers' GitHub repositories [62–64]. Scores that carry a (6) superscript represent performances derived from evaluating the entire dataset.

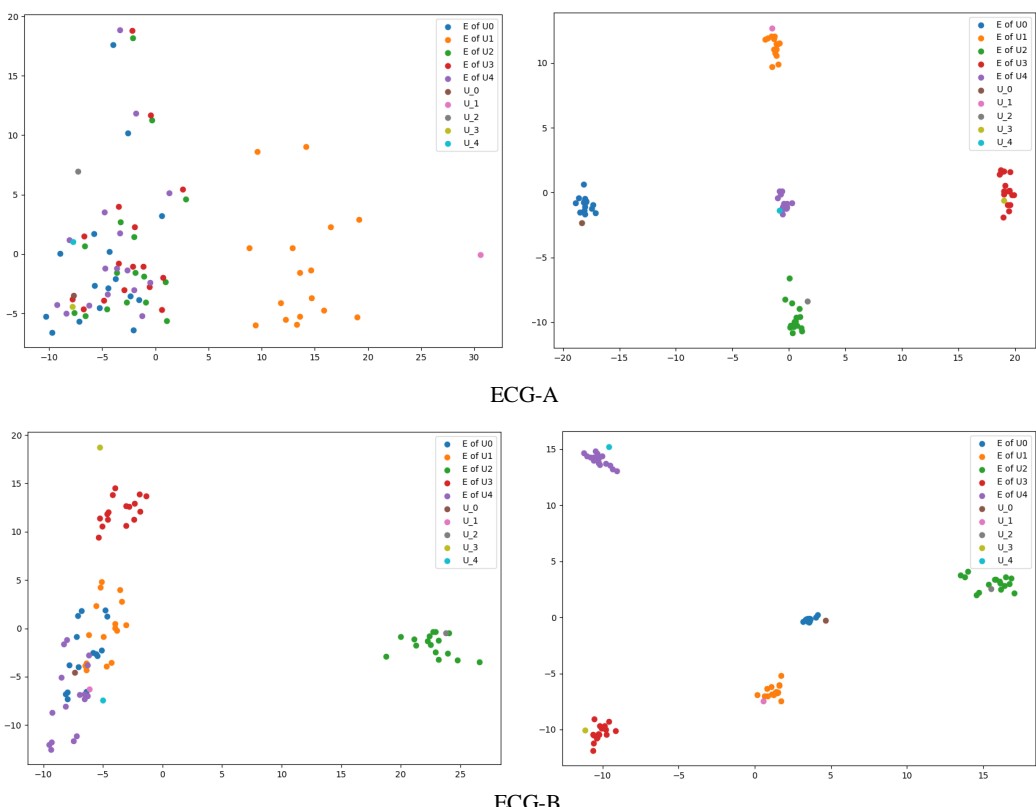

ECG-A

ECG-B

**Figure 14.** Representation distributions before and after training for two sub-time series within the ECG benchmark. **Left**: Pre-training; **Right**: Post-training.

**Table 3.** Performances of our method on the benchmark datasets.

| Datasets | Performance ($F_1$) | | |
|---|---|---|---|
| | Precision | Recall | $F_1$ |
| Power demand | 100 | 96.12 | 98.02 |
| UCR | 100 | 95.335 | 97.57 |
| ECG | 100 | 96.825 | 98.35 |
| 2D-Gesture | 100 | 99.86 | 99.93 |
| PSM | 92.94 | 98.73 | 95.74 |
| SMD | 90.975 | 77.61 | 81.835 |
| MSL | 74.07 | 97.08 | 84.03 |
| SWaT | 71.37 | 97.84 | 82.53 |
| WADI | 88.00 | 22.43 | 39.71 |

As seen in Table 4, our proposed method outperformed others in five out of the nine benchmark datasets. It exhibited superior performance over both univariate datasets. When examining multivariate datasets, our method took the lead in the 2D-Gesture, SMD, and MSL datasets.

For the UCR dataset, which comprises four sub-datasets, detailed results of our method can be found in Table A1 in Appendix B. The ECG dataset consists of six sub-datasets and the performances on these sub-datasets are available in Table A2 in Appendix B. Regarding the SMD dataset, all the performance scores in Table 4 were evaluated on the sub-dataset 'machine1-1' of the SMD dataset. For the performance of our method on the entire SMD dataset, refer to Table A3 in Appendix B. For the MSL dataset, the performance was evaluated on its sub-dataset 'C-1'. Finally, for the SWaT dataset, there are two versions

of the training datasets as shown in Table 5. We tested our method on both versions, resulting in similar performances across both.

**Table 4.** Performance comparison on the benchmark datasets (Performance score: $F_1$).

| Methods | Benchmark Datasets | | | | | | | | |
|---|---|---|---|---|---|---|---|---|---|
| | Power Demand | UCR | ECG | 2D-Gesture | PSM | SMD | MSL | SWaT | WADI |
| MAD-GAN [32] | 65.75 [5] | 91.65 [4] | 48.81 [5] | 42.47 [2] | 36.17 [5] | 91.50 [4] | 91.69 [4] | 77.00 [1] | 37.00 [1] |
| DAGMM [22] | 65.72 [5] | 68.90 [4] | 82.35 [5] | 38.91 [2] | 80.08 [3] | 94.91 [4] | 84.82 [4] | 81.28 [4] | 14.12 [4] |
| MSCRED [36] | 42.50 [5] | 69.76 [4] | 85.37 [5] | 60.17 [2] | 84.64 [5] | 84.14 [4] | 93.63 [4] | 80.72 [4] | 37.41 [4] |
| CAE-M [28] | 95.52 [5] | 82.22 [4] | 62.32 [5] | 97.94 [5] | 72.04 [5] | 93.67 [4] | 87.33 [4] | 81.01 [4] | 41.17 [4] |
| OmniAnomaly [30] | 58.71 [5] | 90.98 [4] | 78.21 [5] | 41.11 [2] | 80.83 [3] | 94.01 [4] | 87.65 [4] | 81.31 [4] | 42.60 [4] |
| TranAD [43] | 49.96 [5] | 96.94 [4] | 33.33 [5] | 90.34 [5] | 73.86 [5] | 96.05 [1] | 94.94 [1] | 81.51 [1] | 49.51 [1] |
| GDN [29] | 95.70 [5] | 81.58 [4] | 77.68 [5] | 91.68 [5] | 72.92 [5] | 83.42 [4] | **95.91** [4] | 81.00 [1] | 57.00 [1] |
| Anomaly Transformer [42] | 96.06 [5] | 79.29 [5] | 94.47 [5] | 98.47 [5] | **97.89** [1] | 92.33 [1,6] | 93.59 [1,6] | **94.07** [1] | **69.24** |
| MTAD-GAT [38] | 58.78 [5] | 87.61 [4] | 79.19 [5] | 88.87 [5] | 82.06 [5] | 86.83 [4] | 87.68 [4] | 81.01 [1] | 41.69 [4] |
| USAD [39] | 95.70 [5] | 94.476 [4] | 62.33 [5] | 93.36 [5] | 73.86 [5] | 94.95 [4] | 88.22 [4] | 84.00 [1] | 42.96 [1] |
| **CL-TAD** | **98.02** | **97.57** | **97.00** | **99.93** | 95.74 | **99.62** | 95.38 | 82.53 | 39.71 |

[1]: indicates the results are referred from the original papers; [2]: indicates the results are referred from [21]; [3]: indicates the results are referred from [42]; [4]: indicates the results are referred from [43]; [5]: indicates the results are obtained from our self-assessment; [6]: For SMD and MSL: [6] indicates the results that are evaluated on the whole dataset, the remaining results are evaluated on the subset, as mentioned in [60].

**Table 5.** Performance of our model on the SWaT dataset.

| Training Option | | | Performance | | |
|---|---|---|---|---|---|
| Version | Training Dataset | Testing Dataset | Precision | Recall | $F_1$ |
| Version 1 | Normal_v.0 | Attack_v.1 | 74.00 | 92.76 | 82.32 |
| Version 2 | Normal_v.1 | Attack_v.1 | 71.37 | 97.84 | 82.53 |

The experiments on the nine benchmark datasets showed that the CL-TAD outperformed other compared methods on five datasets, and achieved competitive performance on two datasets. It did not achieve excellent performance on high-dimensional multivariate time series data like WADI. The CL-TAD produces effective representations on a feature space by incorporating the transformation module and the TCN-based encoder into its encoding process and making use of the contrastive-learning-based training. For high-dimensional time series data like WADI of dimension 127, it appears to struggle in deriving appropriate representations for anomaly detection.

### 4.4. Ablation Study

We undertook additional experiments to better understand the characteristics of our method. Specifically, we tried to answer the following three questions: (**Q1**) Is our method efficient on a limited-sized dataset? (**Q2**) How efficient is our method with different transformation functions? (**Q3**) Is a TCN-based architecture the right choice for feature extraction in our method?

### 4.4.1. **Q1**: Is Our Method Efficient on a Limited-Sized Dataset?

To ascertain the performance of our method on constrained datasets, we undertook evaluations using just 20% of the available training data for two benchmarks: 2D-Gesture and ECG. The results were promising. As presented in Table 6, our method delivered an $F_1$ score of 99.93 on the 2D-Gesture benchmark and 90.64 on the ECG benchmark. Such high scores, despite limited training data, suggest that our approach is resilient to data scarcity and can be effective in real-world scenarios with limited training datasets.

**Table 6.** Performance on the 2D-Gesture and ECG benchmarks with 20% of the available training data.

| Datasets | Performance | | |
| --- | --- | --- | --- |
| | Precision | Recall | $F_1$ |
| 2D-Gesture | 100 | 99.86 | 99.93 |
| ECG | 100 | 85.27 | 90.64 |

### 4.4.2. **Q2**: How Efficient Is Our Method with Different Transformation Functions?

To explore the adaptability of our model to various transformation functions, we assessed its performance on the 2D-Gesture benchmark using transformation functions other than a linear transformation. As shown in Table 7, our model impressively achieved $F_1$ scores of 99.66 and 99.79 when leveraging a singular 1D-CNN and a TCN, respectively, as the transformation functions. This underscores the flexibility and efficiency of our approach across diverse transformation functions. Furthermore, Figure 15 visually compares the accuracy across different transformation functions, further highlighting the performance of a linear transformation in our model.

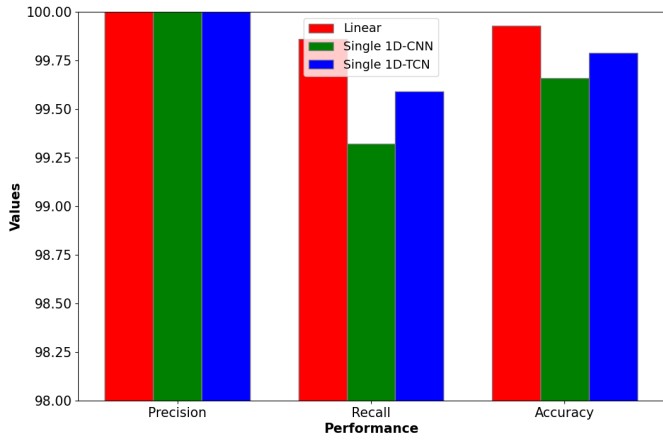

**Figure 15.** Performance of our model on 2D-Gesture benchmark with different transformation functions.

**Table 7.** Performance on 2D-Gesture with different transformation functions.

| Transformation Function | Performance | | |
| --- | --- | --- | --- |
| | Precision | Recall | $F_1$ |
| Single 1D-CNN | 100 | 99.32 | 99.66 |
| Single TCN | 100 | 99.59 | 99.79 |

### 4.4.3. **Q3**: Is aTCN-based Architecture the Right Choice for Feature Extraction in Our Model?

To explore the effectiveness of different architectures in the encoders of our model, we employed a six-layer 1D-CNN as a substitute for our standard setup. The results, detailed in Figure 16 and Table 8, show that the TCN-based encoders consistently outperformed

the 1D-CNN-based counterparts, achieving higher $F_1$ scores on datasets like 2D-Gesture and PSM. Such findings suggest that the 1D-TCN architecture is more robust than the 1D-CNN architecture in the proposed method when it comes to temporal feature extraction for anomaly detection.

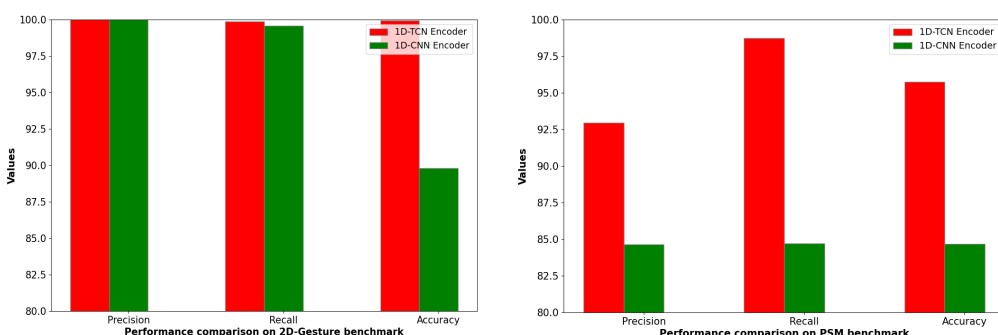

**Figure 16.** Performance of our model on 2D-Gesture and PSM benchmarks with different encoders.

**Table 8.** Performance of our model with a six-Layer 1D-CNN used in Encoders on 2D-Gesture and PSM benchmark datasets.

| Encoder Module | Dataset | Performance | | |
| --- | --- | --- | --- | --- |
| | | Precision | Recall | $F_1$ |
| 6Layer 1D-CNN | 2D-Gesture | 100 | 99.59 | 99.79 |
| | PSM | 84.64 | 84.71 | 84.68 |

## 5. Conclusions

The time series anomaly detection task has recently gained increasing attention in diverse sectors including manufacturing, banking, and healthcare. In this paper, we introduced CL-TAD, a novel contrastive-learning-based method for time series anomaly detection. In our method, we augment the normal data with a combination of a masking technique and a TCN-based encoder–decoder module. Subsequently, a representation generation network is trained using a contrastive learning technique for both the original normal samples and the augmented samples. The training process relies solely on normal time series data without any anomaly events. This characteristic is particularly beneficial for anomaly detection, with the challenges associated with collecting datasets rich in anomalies. For the determination of anomaly score thresholds, we deploy an approach that searches for the value that maximizes the $F_1$ score for a designated held-out dataset. The CL-TAD is potentially quite useful in industries such as manufacturing, finance, and healthcare for which it is time-consuming and expensive to label training time series data because the CL-TAD does not require labeled data. To evaluate the performance of our method, we applied it to the nine well-known benchmark datasets. It outperformed other approaches in five out of the nine benchmarks in terms of $F_1$ performance. In the experiments, the CL-TAD achieved excellent performances on univariate time series data as well as multivariate time series data of which the number of dimensions is not high. The experiments showed that the effective representation could be found for time series data anomaly detection by the proposed method only with normal data. It is practically useful in practical applications for which it is hard to obtain labeled training data.

The reliance on normal data during the training phase renders our approach highly applicable to situations with a scarcity of labeled data. Moreover, our empirical evaluations demonstrated that CL-TAD maintained its robust performance even with limited training data. This means that CL-TAD can be used as a valuable tool for time series anomaly detection, showcasing its potential to address real-world challenges effectively.

**Author Contributions:** Conceptualization, H.C.V.N. and K.M.-L.; methodology, H.C.V.N.; software, H.C.V.N.; validation, H.C.V.N. and K.M.-L.; formal analysis, K.M.L.; investigation, K.M.L.; resources, K.M.L.; data curation, H.C.V.N.; writing—original draft preparation, H.C.V.N.; writing—review and editing, K.M.-L.; visualization, H.C.V.N.; supervision, K.M.-L.; project administration, K.M.-L.; funding acquisition, K.M.-L. All authors have read and agreed to the published version of the manuscript.

**Funding:** This work was supported by the National Research Foundation of Korea (NRF) grant funded by the Korea government (MSIT) (No. 2022R1A5A8026986) and the MSIT (Ministry of Science and ICT), Korea, under the Grand Information Technology Research Center support program (IITP-2023-2020-0-01462) supervised by the IITP (Institute for Information & communications Technology Planning & Evaluation).

**Informed Consent Statement:** Not applicable.

**Data Availability Statement:** The source codes of the proposed method, along with the trained models for the benchmark datasets are available at https://github.com/nguhcv/cl-tad (accessed on 1 August 2023).

**Conflicts of Interest:** The authors declare no conflict of interest.

## Abbreviations

The following abbreviations are used in this manuscript:

| | |
|---|---|
| CL-TAD | Contrastive-learning-based Method for Times-Series Anomaly Detection |
| CNN | Convolutional Neural Network |
| DL | Deep Learning |
| SA | Signal Analysis |
| TCN | Temporal Neural Network |

## Appendix A. Algorithms in the Proposed Method

---

**Algorithm A1:** Masked data batch generation with $\gamma$

---

1 **Input** a batch of subsequences: $\{B_i\}_{i=0}^{S-1}$; length of a subsequence: $L$
2 **Output** a batch of masked subsequences : $\{M_j\}_{j=0}^{N-1}$ $\qquad\qquad\qquad \triangleright N = S \times L$
3 **begin**
4 $\quad$ **for** *i in range* $(0, S-1)$ **do**
5 $\quad\quad$ **for** *j in range* $(iL, iL + L - 1)$ **do**
6 $\quad\quad\quad$ $M_j = copy(B_i)$
7 $\quad\quad\quad$ $M_j[j - iL] = m$ $\qquad\qquad\qquad\qquad \triangleright m$: masking value
8 $\quad\quad$ **end**
9 $\quad$ **end**
10 **end**

---

**Algorithm A2:** Reconstruction loss computation for the reconstruction module $\Phi$

---

1 **Input** a batch of masked samples $\{M_j\}_{j=0}^{N-1}$; Reconstruction module $\Phi$
2 **Output** a batch of reconstructed samples $\{R_j\}_{j=0}^{N-1}$
3 **begin**
4 $\quad$ # *Forward pass*
5 $\quad$ $\{R_j\}_{j=0}^{N-1} = \Phi(\{M_j\}_{j=0}^{N-1})$
6 $\quad$ # *Minimize loss*
7 $\quad$ $L^{rc} = \frac{1}{N} \sum_{j=0}^{N-1} \frac{1}{L} \sum_{t=0}^{L-1} ||B_{\lfloor j/L \rfloor}[t] - R_j[t]||$
8 **end**

---

---

**Algorithm A3:** An update step in the contrastive-learning-based representation learning

---

1    **Input** a batch of subsequence samples: $\{B_i\}_{i=0}^{S-1}$; masking module: $\gamma$; reconstruction module: $\Phi$; transformation module: $\psi$; encoder module: $\Omega$; similarity score hyperparameter: $\tau$; length of a sample: $L$

2    **Output** $\{U_i\}_{i=0}^{S-1}$, $\{E_j\}_{j=0}^{N-1}$

3    **begin**

4      *# Forward pass*

5      *#Generate masking data batch $\{M_j\}_{j=0}^{N-1}$ with module $\gamma$*

6      $\{M_j\}_{j=0}^{N-1} = \gamma\,(\,\{B_i\}_{i=0}^{S-1}\,)$

7      *# Generate positive samples $\{R_j\}_{j=0}^{N-1}$ for $\{B_i\}_{i=0}^{S-1}$ with reconstruction module $\Phi$*

8      $\{R_j\}_{j=0}^{N-1} = \Phi\,(\,\{M_j\}_{j=0}^{N-1}\,)$

9      *# Generate learnable augmented samples $\{T_j\}_{j=0}^{N-1}$ of $\{R_j\}_{j=0}^{N-1}$ with module $\psi$*

10      $\{T_j\}_{j=0}^{N-1} = \psi\,(\,\{R_j\}_{j=0}^{N-1}\,)$

11      *# Captures information from $\{T_j\}_{j=0}^{N-1}$ and map to a new space $\{E_j\}_{j=0}^{N-1}$ for contrastive learning with module $\Omega$*

12      $\{E_j\}_{j=0}^{N-1} = \Omega\,(\,\{T_j\}_{j=0}^{N-1}\,)$

13      *# Captures the information from $\{B_j\}_{j=0}^{N-1}$ and map to a new space $\{U_j\}_{j=0}^{N-1}$ for contrastive learning with module $\Omega$*

14      $\{U_i\}_{i=0}^{S-1} = \Omega\,(\,\{B_i\}_{i=0}^{S-1}\,)$

15      *# Calculate the reconstruction loss $L^{rc}$ based on $\{B_i\}_{i=0}^{S-1}$ and $\{R_j\}_{j=0}^{N-1}$*

16      $L^{rc} = \frac{1}{N}\sum_{j=0}^{N-1}\frac{1}{L}\sum_{t=0}^{L-1}||B_{\lfloor j/L\rfloor}[t] - R_j[t]||$

17      *# Calculate the contrastive loss $L^{ct}$ based on $\{U_i\}_{i=0}^{S-1}$ and $\{E_j\}_{j=0}^{N-1}$ (line 18 ~ 31)*

18      *# Calculate pairwise similarity score between all positive (U,E) pairs based on Equation 4*

19      **for** $i \in (0,S\text{-}1)$ **do**

20        **for** $k \in (iL,\, iL+L\text{-}1)$ **do**

21          $v(U_i, E_k) = \exp\!\left(\frac{U_i^T E_k}{\|U_i\|\|E_k\|} * \frac{\sigma(u(U_i))}{\tau}\right)$       $\triangleright\ \sigma$: sigmoid function

22        **end**

23      **end**

24      *# Calculate pairwise similarity score between all positive (E,U) pairs based on Equation 4*

25      **for** $j \in (0,N\text{-}1)$ **do**

26        $v(E_j, U_{\lfloor j/L\rfloor}) = \exp\!\left(\frac{E_j^T U_{\lfloor j/L\rfloor}}{\|E_j\|\|U_{\lfloor j/L\rfloor}\|} * \frac{\sigma(u(E_j))}{\tau}\right)$       $\triangleright\ \sigma$: sigmoid function

27      **end**

28      **define** $L^{ct}(U_i, E_k) = -log\dfrac{v(U_i,E_k)}{\sum_{j=0}^{N-1}1_{[j\notin[iL,iL+L-1]]}v(U_i,E_j)+\sum_{m=0}^{S-1}1_{[m\neq i]}v(U_i,U_m)+v(U_i,E_k)}$

29      **define** $L^{ct}(E_j, U_{\lfloor j/L\rfloor}) = -log\dfrac{v(E_j,U_{\lfloor j/L\rfloor})}{\sum_{k=0}^{N-1}1_{[k\notin[hL,hL+L-1]]}v(E_j,E_k)+\sum_{i=0}^{S-1}v(E_j,U_i)}$

30      *# Calculate an average contrastive loss*

31      $L^{ct} = \frac{1}{2}\big(\big[\frac{1}{N}\sum_{i=0}^{S-1}\sum_{k=iL}^{iL+L-1}L^{ct}(U_i,E_k)\big] + \big[\frac{1}{N}\sum_{j=0}^{N-1}L^{ct}(E_j,U_{\lfloor j/L\rfloor})\big]\big)$

32      *# Calculate the final loss $L^{final}$*

33      $L^{final} = L^{rc} + L^{ct}$

34      Update weights in modules $\Phi, \psi, \Omega$ to minimize the loss $L^{final}$

35    **end**

---

**Algorithm A4:** Anomaly score computation and anomaly detection

---

**1 Input** A set of subsequences for a testing time series: **W**, masking module: $\gamma$, transformation function: $\psi$, encoder function: $\Omega$, threshold: $\delta$

**2 begin**

**3**   **foreach** *sliding window* $W_i \in \mathbf{W}$ **do**

**4**    **#** *Forward pass*

**5**    #*Generate L masking samples* $\{M_j\}_{j=0}^{L-1}$ *with module* $\gamma$

**6**    $\{M_j\}_{j=0}^{L-1} = \gamma\,(\,W_i)$

**7**    # *Generate L positive samples* $\{R_j\}_{j=0}^{L-1}$ *for* $W_i$ *with reconstruction module* $\Phi$

**8**    $\{R_j\}_{j=0}^{L-1} = \Phi\,(\,\{M_j\}_{j=0}^{L-1})$

**9**    # *Generate L learnable augmented samples* $\{T_j\}_{j=0}^{L-1}$ *of* $\{R_j\}_{j=0}^{L-1}$ *with module* $\psi$

**10**    $\{T_j\}_{j=0}^{L-1} = \psi\,(\,\{R_j\}_{j=0}^{L-1})$

**11**    # *Captures information from* $\{T_j\}_{j=0}^{L-1}$ *and map to a new space* $\{E_j\}_{j=0}^{L-1}$ *with module* $\Omega$

**12**    $\{E_j\}_{j=0}^{L-1} = \Omega\,(\,\{T_j\}_{j=0}^{L-1})$

**13**    # *Captures the information from* $W_i$ *and map to a new space* $U_i$ *with module* $\Omega$

**14**    $U_i = \Omega\,(\,W_i)$

**15**    # *Detect anomaly of* $W_i$ *(line 16 $\sim$ 28)*

**16**    **if** $i = 0$ **then**

**17**     # *Detect anomaly at every time point of* $W_i$

**18**     **for** $j$ *in range* $(0, L-1)$ **do**

**19**      $D(U_i, E_j) = \|U_i/\|U_i\| - E_j/\|E_j\|\|_2$

**20**      **if** $D(U_i, E_j) > \delta$ **then**

**21**       $W_i[j]$ is an anomaly point

**22**      **end**

**23**     **end**

**24**    **else**

**25**     **if** $i > 0$ **then**

**26**      # *Detect anomaly at the last time point of* $W_i$

**27**      # Calculate distance between $U_i$ and $E_{L-1}$

**28**      $D(U_i, E_{L-1}) = \|U_i/\|U_i\| - E_{L-1}/\|E_{L-1}\|\|_2$

**29**      **if** $D(U_i, E_{L-1}) > \delta$ **then**

**30**       $W_i[-1]$ is an anomaly point

**31**      **end**

**32**     **end**

**33**    **end**

**34**   **end**

**35 end**

---

---

**Algorithm A5:** Determination of the list of candidate thresholds

---

1 **Input** A set of subsequences for a test time series: **W**, masking module: $\gamma$,
　transformation module: $\psi$, encoder module: $\Omega$

2 **Output** a list of threshold $\Delta$

3 **begin**

4 　# *Determine the maximum distance between a positive (U,E) pair (line 5 $\sim$ 28)*

5 　$d_{max} = 0$　　　　　　$\triangleright$ $d_{max}$ : maximum distance between a positive (U,E) pair

6 　**foreach** *input data batch* $\{B_i\}_{i=0}^{S-1} \in W$ **do**

7 　　# *Forward pass (line 8 $\sim$ 17)*

8 　　#*Generate masking data batch* $\{M_j\}_{j=0}^{N-1}$ *with module* $\gamma$

9 　　$\{M_j\}_{j=0}^{N-1} = \gamma\,(\,\{B_i\}_{i=0}^{S-1}\,)$

10 　　# *Generate positive samples* $\{R_j\}_{j=0}^{N-1}$ *for* $\{B_i\}_{i=0}^{S-1}$ *with reconstruction module* $\Phi$

11 　　$\{R_j\}_{j=0}^{N-1} = \Phi\,(\,\{M_j\}_{j=0}^{N-1}\,)$

12 　　# *Generate learnable augmented samples* $\{T_j\}_{j=0}^{N-1}$ *of* $\{R_j\}_{j=0}^{N-1}$ *with module* $\psi$

13 　　$\{T_j\}_{j=0}^{N-1} = \psi\,(\,\{R_j\}_{j=0}^{N-1}\,)$

14 　　# *Captures information from* $\{T_j\}_{j=0}^{N-1}$*and map to a new space* $\{E_j\}_{j=0}^{N-1}$ *for*
　　*contrastive learning with module* $\Omega$

15 　　$\{E_j\}_{j=0}^{N-1} = \Omega\,(\,\{T_j\}_{j=0}^{N-1}\,)$

16 　　# *Captures the information from* $\{B_j\}_{j=0}^{N-1}$*and map to a new space* $\{U_j\}_{j=0}^{N-1}$ *for*
　　*contrastive learning with module* $\Omega$

17 　　$\{U_i\}_{i=0}^{S-1} = \Omega\,(\,\{B_i\}_{i=0}^{S-1}\,)$

18 　　# *Determine the maximum distance* $d_{max}$ *(line 19 $\sim$ 28)*

19 　　**for** *i in range (0, S-1)* **do**

20 　　　distances =[]

21 　　　**for** *k in range (iL, iL+L-1)* **do**

22 　　　　# Calculate distance between $U_i$ and $E_k$

23 　　　　$D(U_i, E_k) = \|U_i - E_k\|_2$

24 　　　　distances.append( $D(U_i, E_k)$)

25 　　　**end**

26 　　　$d_{max} = max(d_{max}, max(distances))$

27 　　**end**

28 　**end**

29 　# *Generate a list of thresholds for testing* $d_{max}$ *(line 30 $\sim$ 35)*

30 　$\Delta = []$

31 　$interval = \frac{d_{max}}{1001}$

32 　**for** *c in range (0,1000)* **do**

33 　　$\Delta.append((c + 1) * interval)$

34 　**end**

35 **end**

---

---

**Algorithm A6:** The evaluation of the best $F_1$-based performance

---

1　**Input** A set of subsequences for a test time series: **W**; masking module: $\gamma$; reconstruction module: $\psi$; encoder module: $\Omega$; a list of candidate thresholds: $\Delta$; ground true labels: $Gt$

2　**Output** The best $F_1$ score and its threshold

3　**begin**

4　　　$F_{1-best} = 0; Precision_f = 0; Recall_f = 0; threshold_f = 0;$

5　　　**foreach** $\delta \in \Delta$ **do**

6　　　　　$Os = []$　　**# *Initial empty output sequence Os***

7　　　　　**foreach** *a subsequence* $W_i \in$ **W** **do**

8　　　　　　　**# *Forward pass***

9　　　　　　　#*Generate L masking samples* $\{M_j\}_{j=0}^{L-1}$ *with module* $\gamma$

10　　　　　　$\{M_j\}_{j=0}^{L-1} = \gamma(W_i)$

11　　　　　　# *Generate L positive samples* $\{R_j\}_{j=0}^{L-1}$ *for* $W_i$ *with reconstruction module* $\Phi$

12　　　　　　$\{R_j\}_{j=0}^{L-1} = \Phi(\{M_j\}_{j=0}^{L-1})$

13　　　　　　# *Generate L learnable augmented samples* $\{T_j\}_{j=0}^{L-1}$ *of* $\{R_j\}_{j=0}^{L-1}$ *with module* $\psi$

14　　　　　　$\{T_j\}_{j=0}^{L-1} = \psi(\{R_j\}_{j=0}^{L-1})$

15　　　　　　# *Captures information from* $\{T_j\}_{j=0}^{L-1}$ *and map to a new space* $\{E_j\}_{j=0}^{L-1}$ *with module* $\Omega$

16　　　　　　$\{E_j\}_{j=0}^{L-1} = \Omega(\{T_j\}_{j=0}^{L-1})$

17　　　　　　# *Captures the information from* $W_i$ *and map to a new space* $U_i$ *with module* $\Omega$

18　　　　　　$U_i = \Omega(W_i)$

19　　　　　　**# *Detect anomaly of*** $W_i$

20　　　　　　**if** $i = 0$ **then**

21　　　　　　　　# *Detect anomaly at every time point of* $W_i$

22　　　　　　　　**for** $j$ *in range* $(0, L-1)$ **do**

23　　　　　　　　　　$D(U_i, E_j) = \|U_i - E_j\|_2$

24　　　　　　　　　　**if** $D(U_i, E_j) \leq \delta$ **then**

25　　　　　　　　　　　　$Os.append(0)$

26　　　　　　　　　　**else**

27　　　　　　　　　　　　$Os.append(1)$

28　　　　　　　　　　**end**

29　　　　　　　　**end**

30　　　　　　**else**

31　　　　　　　　**if** $i > 0$ **then**

32　　　　　　　　　　# *Detect anomaly at the last time point of* $W_i$

33　　　　　　　　　　# *Calculate distance between* $U_i$ *and* $E_{L-1}$

34　　　　　　　　　　$D(U_i, E_{L-1}) = \|U_i/\|U_i\| - E_{L-1}/\|E_{L-1}\|\|_2$

35　　　　　　　　　　**if** $D(U_i, E_j) \leq \delta$ **then**

36　　　　　　　　　　　　$Os.append(0)$

37　　　　　　　　　　**else**

38　　　　　　　　　　　　$Os.append(1)$

39　　　　　　　　　　**end**

40　　　　　　　　**end**

41　　　　　　**end**

42　　　　　**end**

43　　　　　point-adjustment $(Os, Gt)$

44　　　　　Calculate $F_1, Precision, Recall$ $(Os, Gt)$

45　　　**end**

46　　　**if** $F_{1-best} < F_1$ **then**

47　　　　　$F_{1-best} = F_1; Precision_f = Precision; Recall_f = Recall; threshold_f = \delta$

48　　　**end**

49　　　**return** $F_{1-best}; Precision_f; Recall_f; threshold_f$

50　**end**

---

---

**Algorithm A7:** Normalization of the dataset

---

1 **Input** original training time series: $X_{train}$; original test time series: $X_{test}$;
  dimension of data point: $d$
2 **begin**
3     # Calculate maximum value $v_{max}$ and minimum value $v_{min}$ on $X_{train}$
4     $v_{max} = max(X_{train})$
5     $v_{min} = min(X_{train})$
6     # Normalize values on both $X_{train}$ and $X_{test}$
7     **for** *i in range* $(0, d-1)$ **do**
8         $X_{train}[i][:] = (X_{train}[i][:] - v_{min}/(v_{max} - v_{min}))$
          $X_{test}[i][:] = (X_{test}[i][:] - v_{min}/(v_{max} - v_{min}))$
9     **end**
10 **end**

---

## Appendix B. Detailed Performance of Our Method on the Sub-Datasets of Some Benchmark Datasets

**Table A1.** Detailed Performance on the UCR benchmark.

| UCR Sub-Time Series | Performance | | |
|---|---|---|---|
| | Precision | Recall | $F_1$ |
| 135_ | 100 | 92.31 | 96.00 |
| 136_ | 100 | 99.10 | 99.55 |
| 137_ | 100 | 99.03 | 99.51 |
| 138_ | 100 | 90.90 | 95.23 |
| **Average** | 100 | 95.335 | 97.57 |

**Table A2.** Detailed Performance on the ECG benchmark.

| ECG Sub-Time Series | Performance | | |
|---|---|---|---|
| | Precision | Recall | $F_1$ |
| ECG-A chfdb_chf01_275 | 100 | 99.62 | 99.81 |
| ECG-B chfdb_chf13_45590 | 100 | 99.37 | 99.68 |
| ECG-C chfdbchf15 | 100 | 79.68 | 88.69 |
| ECG-D ltstdb_20221_43 | 100 | 90.20 | 94.85 |
| ECG-E ltstdb_20321_240 | 100 | 98.58 | 99.28 |
| ECG-F mitdb_100_180 | 100 | 99.47 | 98.35 |
| Average | 100 | 94.48 | 97.00 |

**Table A3.** Detailed Performance on the SMD benchmark.

| SMD Sub-Time Series | Performance | | |
| --- | --- | --- | --- |
| | Precision | Recall | $F_1$ |
| machine1-1 | 99.51 | 99.74 | 99.62 |
| machine2-1 | 80.13 | 98.28 | 88.29 |
| machine3-2 | 98.09 | 64.40 | 77.76 |
| machine3-7 | 86.17 | 48.01 | 61.67 |
| Average | 90.975 | 77.61 | 81.835 |

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
