# Peer review of "CL-TAD: A Contrastive-Learning-Based Method for Time Series Anomaly Detection"

_applsci, doi:10.3390/app132111938_

Round 1

Reviewer 1 Report

Comments and Suggestions for Authors

The article introduces CL-TAD (Contrastive Learning-based method for Time Series Anomaly Detection), a novel approach to addressing the challenges of anomaly detection in time series data. By leveraging contrastive learning techniques and a unique positive sample generation process, CL-TAD showcases promising results on benchmark datasets, outperforming several state-of-the-art methods.

Recommendations to Strengthen the Article:

In-Depth Result Analysis: Extend the discussion of experimental results to include an analysis of why CL-TAD outperforms other methods on specific datasets. Consider providing visualizations or examples to illustrate the strengths of the approach.

Practical Applications: Emphasize the practical applications and potential impact of CL-TAD in industries such as manufacturing, finance, and healthcare. Explain how the method can address real-world challenges effectively.

Code and Resources Accessibility: Ensure that the availability of code and resources is prominently featured, with clear and accessible links to GitHub repositories and comprehensive documentation for implementation and reproduction of results.

Conciseness and Clarity: While maintaining technical depth, aim for a more concise and well-organized article structure. Eliminate unnecessary details and focus on key insights to make the article more reader-friendly.

Conclusion Enhancement: In the conclusion section, reiterate the key contributions and advantages of CL-TAD, emphasizing its potential to solve critical problems in time series anomaly detection.

By implementing these recommendations, the article can offer a more comprehensive and accessible presentation of the CL-TAD approach and its contributions to the field of time series anomaly detection.

Author Response

Dear Reviewer,

Thanks for your time and valuable comments.
We, the authors, have done our best to improve the manuscript according to all the comments.
Best regards,

Keon Myung Lee

--------------------------------------------------------------------------------------------------------------------------------------
The article introduces CL-TAD (Contrastive Learning-based method for Time Series Anomaly Detection), a novel approach to addressing the challenges of anomaly detection in time series data. By leveraging contrastive learning techniques and a unique positive sample generation process, CL-TAD showcases promising results on benchmark datasets, outperforming several state-of-the-art methods.

Recommendations to Strengthen the Article:

In-Depth Result Analysis: Extend the discussion of experimental results to include an analysis of why CL-TAD outperforms other methods on specific datasets. Consider providing visualizations or examples to illustrate the strengths of the approach.
---------------------------------------------------------------------------------------------------------------Answer:
In the revised manuscript, we added the following part in Section 4.3.
The experiments on the 9 benchmark datasets showed that the CL-TAD outperformed other compared methods on 5 datasets, achieved competitive performance on 2 datasets. It did not achieved excellent performance on the high-dimensional multivariate time series data like WADI. The CL-TAD produces better representations on a feature space by incorporating the transformation module and the TCN-based encoder into its encoding process, and making use of contrastive learning-based training. For high-dimensional time series data like WADI of dimension 127, it seems not to be good at finding appropriate representations for anomaly detection.

---------------------------------------------------------------------------------------------------------------Practical Applications: Emphasize the practical applications and potential impact of CL-TAD in industries such as manufacturing, finance, and healthcare. Explain how the method can address real-world challenges effectively.
---------------------------------------------------------------------------------------------------------------Answer:
In the revised manuscript, we added the following part in Conclusions.

The CL-TAD is potentially quite useful in industries such as manufacturing, finance, and healthcare for which it is time-consuming and expensive to label training time series data because the CL-TAD does not require labelled data.
In the experiments, the CL-TAD achieved excellent performances on univariate time series data as well as multivariate time series data of which dimension is not high (like more than 50). The CL-TAD is practically useful in practical applications for which it is hard to get labelled training data. 

---------------------------------------------------------------------------------------------------------------Code and Resources Accessibility: Ensure that the availability of code and resources is prominently featured, with clear and accessible links to GitHub repositories and comprehensive documentation for implementation and reproduction of results.
---------------------------------------------------------------------------------------------------------------Answer:
We have set up a GitHub repository at https://github.com/nguhcv/cl-tad and provide its link in the manuscript at the Abstract and the Data Availability Statement section of the revised manuscript.

---------------------------------------------------------------------------------------------------------------Conciseness and Clarity: While maintaining technical depth, aim for a more concise and well-organized article structure. Eliminate unnecessary details and focus on key insights to make the article more reader-friendly.
---------------------------------------------------------------------------------------------------------------
Answer:
We improved the manuscript according to the comments.

---------------------------------------------------------------------------------------------------------------Conclusion Enhancement: In the conclusion section, reiterate the key contributions and advantages of CL-TAD, emphasizing its potential to solve critical problems in time series anomaly detection.
---------------------------------------------------------------------------------------------------------------Answer:
According to the comments we updated the conclusions as following:

The time series anomaly detection task has recently gained increasing attention in diverse sectors including manufacturing, banking, and healthcare. In this paper, we introduced CL-TAD, a novel contrastive learning-based method for time series anomaly detection. In our method, we augment the normal data with a combination of a masking technique and a TCN-based encoder-decoder module. Subsequently, a representation generation network is trained using a contrastive learning technique for both the original normal samples and their augmented ones. The training process relies solely on normal time series data without any anomaly events. This characteristic is particularly beneficial for anomaly detection, with the challenges associated with collecting datasets rich in anomalies. For the determination of anomaly score thresholds, we deploy an approach that searches for the value that maximizes the F_1-score for a designated held-out dataset. The CL-TAD is potentially quite useful in industries such as manufacturing, finance, and healthcare for which it is time-consuming and expensive to label training time series data because the CL-TAD does not require labelled data. To evaluate the performance of our method, we applied it to the nine well-known benchmark datasets. It outperformed others in 5 out of 9 benchmarks in terms of F_1 performance. In the experiments, the CL-TAD achieved excellent performances on univariate time series data as well as multivariate time series data of which dimension is not high. The experiments showed that the effective representation could be found for time series data anomaly detection by the proposed method only with normal data. It is practically useful in practical applications for which it is hard to get labelled training data.

The reliance on normal data during the training phase renders our approach highly applicable to situations with a scarcity of labeled data. Moreover, our empirical evaluations demonstrated that CL-TAD maintained its robust performance even with limited training data. It means that CL-TAD can be used as a valuable tool for time series anomaly detection, showcasing its potential to address real-world challenges effectively.

Reviewer 2 Report

Comments and Suggestions for Authors

The authors present a novel method for anomaly detection in time series data, called CL-TAD (Contrastive Learning-based method for Times series Anomaly Detection). The method uses a contrastive learning-based representation learning technique, has two main components: positive sample generation; and contrastive learning-based representation learning.

The first component generates positive samples (by trying to reconstruct the original data for masked samples), and then, these positive samples along with the original data, are the input for the second component (the contrastive learning-based representation learning component).

The representations of the input original data and their masked data are then used to detect anomalies.

The experimental results demonstrated that the CL-TAD method achieved the best performance on five datasets (out of nine benchmark datasets) over other recent 10 methods.

The presented approach offers a very promising solution for effectively detecting anomalies in time series data.

Globally, the manuscript is very well written and organized. My only recommendation is for the authors to carefully read the whole document and correct some minor English errors/typos; please refer to the attached commented PDF document, where some of the needed corrections are highlighted. Additionally, the authors should also mention the version number of the software used during their experiments, and correct the numbering of tables in the final section of the document.

Comments on the Quality of English Language

Please refer to the comments above.

Author Response

Dear Reviewer,

Thanks for your time and valuable comments.
We, the authors, have done our best to improve the manuscript according to all the comments.
Best regards,

Keon Myung Lee

--------------------------------------------------------------------------------------------------------------------------------------- My only recommendation is for the authors to carefully read the whole document and correct some minor English errors/typos; please refer to the attached commented PDF document, where some of the needed corrections are highlighted. Additionally, the authors should also mention the version number of the software used during their experiments, and correct the numbering of tables in the final section of the document.
---------------------------------------------------------------------------------------------------------------------------------------

Answer:
According to the comments on the attached commented PDF file, we updated all the typos and grammatical mistakes. We added the software version to the used software. The authors greatly appreciate the careful comments on the manuscript. 

Reviewer 3 Report

Comments and Suggestions for Authors

This paper presents a new method for detecting anomalies in time series. This method is introduced in detail and compared to other methods using different datasets. The following suggestions are listed to improve the paper quality further.

1. In addition to the anomaly detection methods in the Introduction, Bayesian methods, such as Anomaly detection of structural health monitoring data using the maximum likelihood estimation-based Bayesian dynamic linear model, are also essential and should be briefly reviewed. 

2. It is better to illustrate that each compared method has achieved its best performance by using the optimal hyperparameters. 

3. The evaluation criteria for the model's performance should be clarified and presented clearly in the corresponding tables.  

4. The fonts in the figures are too small, which should be checked.

Comments on the Quality of English Language

Minor editing of English language required.

Author Response

Dear Reviewer,

Thanks for your time and valuable comments.
We, the authors, have done our best to improve the manuscript according to all the comments.
Best regards,

Keon Myung Lee

---------------------------------------------------------------------------------------------------------------
This paper presents a new method for detecting anomalies in time series. This method is introduced in detail and compared to other methods using different datasets. The following suggestions are listed to improve the paper quality further.

  1. In addition to the anomaly detection methods in the Introduction, Bayesian methods, such as Anomaly detection of structural health monitoring data using the maximum likelihood estimation-based Bayesian dynamic linear model, are also essential and should be briefly reviewed. 
    ---------------------------------------------------------------------------------------------------------------
    Answer
    :
    We added the Bayesian methods as an approach for anomaly detection methods in the section 1. Introduction as follows:
    Various methods have been developed for the task, which can be categorized into five approaches: statistical [5–9], signal analysis (SA) [10–12], system modeling-based[13], machine learning (ML) [ 14 – 19], and deep learning (DL) [20– 43] approaches. The statistical approach involves creating a statistical model by calculating distributions and measures such as mean, variance, median, quantile, and others. The SA-based approach utilizes time-frequency domain analysis techniques like Fourier transform, to identify anomalies. The system modeling-based approach constructs a mathematical model like Bayesian dynamic linear model[13] that simulates the latent process to produce the time series of interest.

    As a reference for them, we added the following one:
    [13]. Wang, Q. A.; Wang, C. B.; Ma, Z. G.; Chen, W.; Ni, Y. Q.; Wang, C. F.; Yang, P.-X.; Guan, P. X. Bayesian dynamic linear model framework for structural health monitoring data forecasting and missing data imputation during typhoon events. Structural Health Monitoring, 2022, 21(6), 2933-2950

---------------------------------------------------------------------------------------------------------------2. It is better to illustrate that each compared method has achieved its best performance by using the optimal hyperparameters. 
---------------------------------------------------------------------------------------------------------------
Answer
:
According to the comment, we mentioned that each compared method has achieved its best performance by using the curated optimal hyperparameters in Section 4.3. In addition, the trained models by our method have be uploaded at GitHub repository https://github.com/nguhcv/cl-tad which is reported in the Data Availability Statement section section.

---------------------------------------------------------------------------------------------------------------3. The evaluation criteria for the model's performance should be clarified and presented clearly in the corresponding tables.  
---------------------------------------------------------------------------------------------------------------
Answer
:
For Table 3 and 4, we added the information that the performance score is F_1.

---------------------------------------------------------------------------------------------------------------4. The fonts in the figures are too small, which should be checked.
---------------------------------------------------------------------------------------------------------------
Answer
:
According to the comments, we extended the figures in the revised manuscript.  
